# AutoGraph-R1: End-to-End Reinforcement Learning for Knowledge Graph Construction

## Abstract

Building effective knowledge graphs (KGs) for Retrieval-Augmented Generation (RAG) is pivotal for advancing question answering (QA) systems. However, its effectiveness is hindered by a fundamental disconnect: the knowledge graph (KG) construction process is decoupled from its downstream application, yielding suboptimal graph structures. To bridge this gap, we introduce AutoGraph-R1, the first framework to directly optimize KG construction for task performance using Reinforcement Learning (RL). AutoGraph-R1 trains an LLM constructor by framing graph generation as a policy learning problem, where the reward is derived from the graph's functional utility in a RAG pipeline. We design two novel, task-aware reward functions, one for graphs as knowledge carriers and another as knowledge indices. Across multiple QA benchmarks, AutoGraph-R1 consistently enables graph RAG methods to achieve significant performance gains over using task-agnostic baseline graphs. Our work shows it is possible to close the loop between construction and application, shifting the paradigm from building intrinsically "good" graphs to building demonstrably "useful" ones.

## 1 Introduction

Retrieval-Augmented Generation (RAG) has become a cornerstone for enhancing Large Language Models (LLMs), enabling them to ground responses in external knowledge, reduce hallucinations, and incorporate real-time, domain-specific information (Gao et al., 2024a; Peng et al., 2024a). A particularly promising frontier is graph-based RAG, where LLMs leverage the structured nature of Knowledge Graphs (KGs) for complex data sensemaking and reasoning (Edge et al., 2025). The typical pipeline begins by constructing a KG from unstructured text, often using LLM-driven extractors and heuristics (Han et al., 2024; Lairgi et al., 2024; Bai et al., 2025), which then supports downstream question answering (Gutiérrez et al., 2025a; Sun et al., 2024).

Despite its potential, the prevailing graph-based RAG paradigm suffers from a fundamental disconnect: the process is split into two isolated phases. First, a construction phase, where a graph is built and evaluated on intrinsic metrics like precision and coverage (Huang et al., 2025a). Second, an application phase, where this static graph is used for a downstream task. The critical flaw in this approach is that a "good" graph by intrinsic standards is not necessarily a "useful" one for the end task (Xue & Zou, 2022). For instance, a graph built to maximize factual accuracy might be structurally fragmented, causing retrievers to fail on multi-hop questions that require connecting distant information, as illustrated in Figure 1.

This disconnect persists because closing the loop between downstream performance and graph construction is technically challenging. The construction process generating discrete (subject, predicate, object) triples and performing entity resolution is inherently non-differentiable. Consequently, standard gradient-based optimization cannot backpropagate performance signals from a downstream task, such as question answering accuracy, to guide the graph generation model. The graph, once built, cannot learn from its failures. To bridge this gap, we employ Reinforcement Learning (RL). While prior work has used RL to refine retrieval over search tools or improve query reformulation (Jin et al., 2025; Jiang et al., 2025b; Luo et al., 2025), our work is the first to leverage RL to directly optimize the KG construction process itself. We introduce AutoGraph-R1, a framework that fine-tunes an LLM-based graph generator by optimizing for downstream task performance. As shown in Figure 2, the graph generation model learns a construction policy from raw text. The utility of the generated

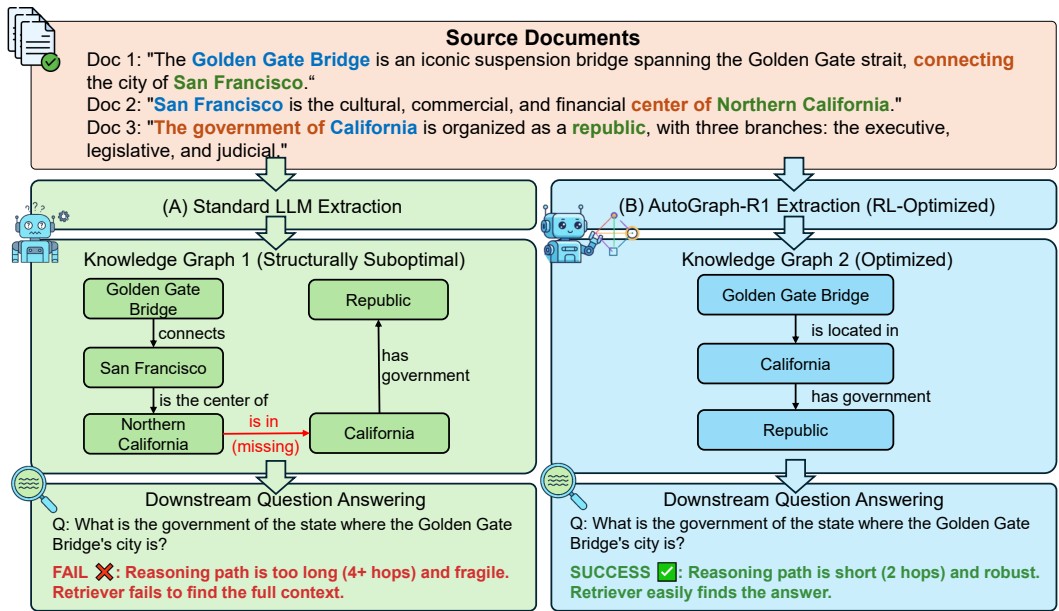

Figure 1: Bridging the Disconnection Between KG Construction and Utility. (A) A KG optimized for intrinsic metrics can be too fragmented for multi-hop QA, causing retrieval to fail. (B) AutoGraph-R1 uses an end-task RL reward to build a functionally superior graph, ensuring its structure is optimized to support various graph retrievers for successful reasoning.

graph is then evaluated in a downstream RAG pipeline, yielding a reward signal. This task-aware reward is used to update the generator's parameters via policy gradient methods, guiding it to produce graphs that are not just factually accurate but functionally optimal, for instance, by creating valid paths that facilitate complex reasoning.

To the best of our knowledge, this is the first work to use reinforcement learning to optimize KG construction for general downstream tasks using LLMs. We focus on complex question answering over benchmark datasets requiring multi-document reasoning (Yang et al., 2018; Trivedi et al., 2022; Mallen et al., 2023). Our framework's reward function is based on the utility of the resulting graph, evaluating whether it serves as an effective knowledge index for retrieving useful text chunks or provides subgraphs that directly support the reasoning process. By designing these task-aware rewards, we directly align the objectives of KG construction with end-task performance.

In summary, our contributions are:

- We introduce AutoGraph-R1, a novel RL framework that directly optimizes knowledge graph construction for downstream utility, bridging the critical gap between graph quality and task performance.

- We design and implement task-aware reward functions that successfully align KG structure with the demands of complex reasoning tasks, compelling the model to build functionally superior graphs.

- Through extensive experiments, we demonstrate that integrating AutoGraph-R1's graphs into a state-of-the-art RAG pipeline yields significant performance gains on multiple QA benchmarks, validating that RL-driven graph construction improves downstream task utility.

## 2 RELATED WORK

### 2.1 GRAPH-BASED RETRIEVAL-AUGMENTED GENERATION

Large Language Models (LLMs), despite demonstrating strong reasoning capabilities (DeepSeek-AI et al., 2025a), remain susceptible to factual hallucinations (Ji et al., 2023; Huang et al., 2025b) and knowledge incompleteness (Peng et al., 2023). Retrieval-Augmented Generation (RAG) (Lewis et al., 2021; Gao et al., 2024b) mitigates these issues by grounding LLMs in external knowledge sources, thereby improving factual accuracy and reasoning. A burgeoning area of research extends RAG with graph-structured knowledge (Peng et al., 2024b; Xiang et al., 2025; Zhang et al., 2025a; Han

et al., 2025). In these pipelines, graphs serve two primary functions. First, as knowledge indices, where the graph organizes and connects raw text chunks, and its structural properties are leveraged for more sophisticated retrieval strategies (Liang et al., 2024; Liu et al., 2024b; Zhang et al., 2024b; Wang et al., 2023; Li et al., 2024a). Methods like HippoRAG (Gutiérrez et al., 2025a;b) exemplify this by exploiting structural connections to access relevant information more effectively. Second, as knowledge carriers, where the graph itself is the primary information source, and the model reasons directly over recovered subgraphs (Shen et al., 2025b; Liu et al., 2024a). This paradigm is adopted by approaches such as Think-on-Graph (Ma et al., 2025; Sun et al., 2024), SubgraphRAG (Li et al., 2025a), StructRAG (Li et al., 2024b), and KnowGPT (Zhang et al., 2024a).

The construction of KGs has evolved from traditional rule-based systems like OpenIE (Angeli et al., 2015) to more flexible LLM-based pipelines such as PiVE (Han et al., 2024), iText2KG (Lairgi et al., 2024), KGGEN (Mo et al., 2025), GraphRAG (Edge et al., 2025), and AutoSchemaKG (Bai et al., 2025). While powerful, these LLM-driven methods typically generate a static graph based on fixed prompts or heuristics, often evaluated using intrinsic metrics. However, the optimal structure of a KG is highly dependent on variety of downstream applications (Gubanov et al., 2024; Wu et al., 2024; Zhao et al., 2024; He et al.; 2024; Liu et al., 2024c). For instance, a graph acting as a text index may prioritize fine-grained partitioning, while one used for reasoning chains requires long-range connectivity (Jin et al., 2024; Huang et al., 2024). This creates the disconnect we identified earlier: a graph built to be "good" in isolation may be functionally poor for a specific task. Our work addresses this gap by optimizing graph construction directly for downstream performance, a problem that, to our knowledge, has not been systematically investigated, despite progress in KG refinement and completion techniques (Chen et al.; 2024b;a; Zhang et al., 2022; Dong et al., 2023).

## 2.2 REINFORCEMENT LEARNING FOR LANGUAGE MODEL OPTIMIZATION

Reinforcement learning (RL) (Kaelbling et al., 1996) offers a powerful framework for optimizing the sequential decision-making capabilities of LLMs by enabling them to learn through environmental interaction and reward feedback (Kaufmann et al., 2024; Xi et al., 2025). As LLMs have become more powerful through fine-tuning, methodological advances—from Reinforcement Learning from Human Feedback (RLHF) (Ouyang et al., 2022) to more scalable and cost-effective algorithms like Proximal Policy Optimization (PPO) (Schulman et al., 2017), Dynamic Sampling Policy Optimization (DAPO) (Yu et al., 2025), and Group Relative Policy Optimization (GRPO) (DeepSeek-AI et al., 2025a)—have enabled successful applications in diverse domains, including open-domain retrieval and scientific discovery. (Zheng et al., 2025; Yu et al., 2024; Zhu et al., 2025; Shen et al., 2025a).

Prior to its widespread adoption for LLM alignment, RL had been explored for knowledge base tasks, such as bidirectional text-to-graph conversion (Dognin et al., 2021). More recently, RL has proven effective in training LLMs to interact with external tools, such as search engines (Jin et al., 2025; Jiang et al., 2025b; Li et al., 2025b; Zhang et al., 2025b; Jiang et al., 2025a). Notably, frameworks like Graph-R1 (Luo et al., 2025) have shown that RL can teach an LLM to effectively navigate graph-structured tools to improve retrieval. However, these works use RL to learn a policy for navigating or querying an existing knowledge source. Our approach is fundamentally different: we use RL to learn a policy for constructing the knowledge source itself. To our knowledge, this is the first application of RL to directly optimize a KG's structure based on its measured utility in a downstream reasoning task. This distinction forms the motivation for AutoGraph-R1.

## 3 PRELIMINARIES

In this section, we formalize the key concepts underlying AutoGraph-R1, including knowledge graph construction, graph-based retrieval, and answer generation within a RAG pipeline.

### 3.1 KNOWLEDGE GRAPH CONSTRUCTION

We define a knowledge graph (KG) as a directed, labeled graph $\mathcal{G} = (\mathcal{V}, \mathcal{E}, \mathcal{R})$, constructed from a set of documents $\mathcal{D}$. Here, $\mathcal{V}$ is the set of nodes, $\mathcal{R}$ the set of relation types, and $\mathcal{E} \subseteq \mathcal{V} \times \mathcal{R} \times \mathcal{V}$ the set of edges represented as triples $(s, r, o)$. Nodes $s, o \in \mathcal{V}$ may correspond to entities, events, or concepts, and $r \in \mathcal{R}$ denotes a relation type. Following prior work (Zhang et al., 2025a), we consider two principal configurations for the graph's role.

**Graphs as Knowledge Carriers** In this configuration, the graph serves as a self-contained knowledge base. It consists of factual triples $(s, r, o) \in \mathcal{E}$, where nodes $s, o \in \mathcal{V}$ are entities and $r \in \mathcal{R}$ is the relation. These triples act as discrete, structured knowledge units that are retrieved and processed directly by the downstream model.

**Graphs for Knowledge Indexing** Alternatively, the graph functions as a structured index over the raw document corpus $\mathcal{D}$. Nodes are augmented with pointers to text spans, denoted $\tau(v)$. Formally, the node set can be partitioned into entity nodes $\mathcal{V}_e$ and document or chunk nodes $\mathcal{V}_d$, such that $\mathcal{V} = \mathcal{V}_e \cup \mathcal{V}_d$. This hybrid structure allows the graph to guide retrieval not only of structured facts but also of the original, unstructured text passages.

### 3.2 Retrieval Module

Given a query $q$, the goal of the retrieval module is to produce a set of evidence units $\mathcal{C}(q)$ that will be passed to the LLM for answer generation. We consider two complementary retrieval strategies.

**Graph Knowledge Retriever** This retriever, denoted $\mathcal{R}_{\text{graph}}$, operates directly on the graph structure. Given a query $q$, it identifies relevant components such as individual triples, multi-hop paths, or entire subgraphs. Formally, we define its output as a set of structured evidence $\mathcal{P}(q) = \mathcal{R}_{\text{graph}}(q, \mathcal{G})$. The elements of $\mathcal{P}(q)$ are then linearized into text to serve as context for the LLM.

**Graph-based Text Retriever** This retriever, denoted $\mathcal{R}_{\text{text}}$, uses the graph as an index to find relevant text passages from the source corpus. It leverages graph connectivity to identify promising document nodes. Formally, $\mathcal{R}_{\text{text}}(q, \mathcal{G}) \mapsto \mathcal{T}(q)$, where $\mathcal{T}(q) \subseteq \{\tau(v) \mid v \in \mathcal{V}_d\}$ is a set of raw text passages linked from document nodes.

### 3.3 Answer Generation

The final answer generation step uses a large language model $\pi^{ans}$ to synthesize an answer $\hat{y}$ from the query $q$ and the retrieved evidence $\mathcal{C}(q)$. The evidence context $\mathcal{C}(q)$ is composed of either the linearized graph structures $\mathcal{P}(q)$ from the graph knowledge retriever or the text passages $\mathcal{T}(q)$ from the graph-based text retriever. The final answer $\hat{y}$ is generated by conditioning the LLM on the query and evidence: $\hat{y} = \pi^{ans}(q, \mathcal{C}(q))$. This unified framework allows our optimization process to apply to both types of graph construction, directly linking the structure of $\mathcal{G}$ to its utility in the final QA-task.

## 4 AutoGraph-R1

### 4.1 Reinforcement Learning for Graph Construction

**AutoGraph-R1**, an end-to-end reinforcement learning (RL) framework that directly optimizes knowledge graph (KG) construction with downstream task performance as the reward signal. The framework unifies two common graph-augmented retrieval paradigms: *Graph RAG* (retrieval over entity triples) and *Graph Text RAG* (retrieval over text nodes through graph index).

As shown in Figure 2, AutoGraph-R1 consists of three components: (1) a KG construction policy model $\pi_\theta^{KG}$, instantiated as a large language model (LLM), which maps a list of documents $\mathcal{D}$ into a graph $\mathcal{G}$; (2) a frozen RAG server with a fixed answer generator $\pi^{ans}$, which retrieves from $\mathcal{G}$ and produces an answer $\hat{y}$ to the input query $q$; (3) a task-specific reward function $R(q, \hat{y}, y, \mathcal{G})$ that evaluates how well the constructed graph supports QA, where $y$ is the gold answer.

**Task-Aware Training Loop** A central design choice in AutoGraph-R1, inspired by s3 (Jiang et al., 2025b), is to freeze the retrieval module while the KG construction policy $\pi_\theta^{KG}$ adapts. During training, each sample $(q, y, \mathcal{D}_q)$—comprising a query, a gold answer, and relevant documents—triggers a full end-to-end loop of KG construction, retrieval, and answer generation. Crucially, the definition of a "useful" graph is contingent on the retrieval paradigm. We therefore tailor the training process for two distinct scenarios, aligning the KG's structure with its intended function.

**Training with a Graph Knowledge Retriever** When the KG acts as a knowledge carrier, retrieval quality is measured by its ability to provide a self-contained, structured context for reasoning. To isolate the impact of graph structure, we employ a simple subgraph retriever. Given a query $q$, we extract its named entities to serve as anchors and retrieve the $n$-hop neighborhood surrounding them to

Figure 2: Overview of the AutoGraph-R1 Framework. AutoGraph-R1 optimizes knowledge graph construction for downstream utility using reinforcement learning. During the training phase (left), a graph constructor is fine-tuned with GRPO. The reward signal is derived from the performance of a graph retriever on the generated KG, directly measuring the graph's functional quality. During the inference phase (right), the trained constructor is used to build a large-scale KG from a general corpus, which then serves a downstream graph-based RAG system.

form the context $\mathcal{P}(q)$. This design intentionally bypasses dense vector similarity, forcing the reward signal to reflect the graph's relational completeness and structural integrity. The policy is rewarded for creating graphs where the correct answer is directly *deducible* from the retrieved subgraph.

***Training with a Graph-based Text Retriever.*** When the KG serves as a knowledge index, its utility is determined by how well it guides the retriever to relevant text passages. We adapt the HippoRAG-2 retriever (Gutiérrez et al., 2025b) for this purpose. First, candidate triples $(s, r, o)$ are selected based on embedding similarity to the query $q$ (using Qwen-3-0.6B). However, unlike the original method, we then use *only* these triple-level similarities to initialize a Personalized PageRank algorithm over the constructed graph $\mathcal{G}$. This process propagates relevance scores through the graph structure, ultimately identifying text nodes that are structurally connected to the most pertinent facts, from which the top-$N$ passages are returned. The policy is therefore incentivized to build graphs where structural connectivity, not just semantic similarity, is a reliable signal for identifying crucial evidence passages, including both direct and complementary information that might otherwise be overlooked.

In both scenarios, by freezing the retriever and aligning the reward with its specific mechanism, AutoGraph-R1 ensures the graph construction policy learns to produce graphs that are functionally optimized for a given downstream retrieval strategy.

## 4.2 REWARD DESIGN FOR FUNCTIONAL GRAPH CONSTRUCTION

A primary challenge in optimizing this end-to-end pipeline is the sparse and indirect nature of the learning signal, where a single reward is given after a long sequence of construction actions. This creates a severe credit assignment problem and places a heavy burden on the quality of the reward signal itself. While the final answer's F1 score has been explored as a reward in prior work (Jin et al., 2025), we find its properties make it a challenging choice for our goal of guiding graph construction. The F1 score is brittle; minor phrasing variations in the LLM output can cause swings in the metric, an issue that persists even with deterministic decoding. This instability results in a noisy reward that can impede or destabilize policy optimization. Our approach is therefore motivated by the need to design more direct and stable, task-specific rewards better suited to our problem.

To overcome these challenges, we design two distinct reward functions that provide a more direct and stable learning signal by measuring the functional utility of the graph for a specific retrieval task.

***Graph Knowledge Retriever*** Retrieval operates on subgraphs or relation paths. The key requirement is that the gold answer $y$ should be *deducible* from $\mathcal{G}$. An answer $y$ is considered deducible from $\mathcal{G}$ if the retrieved triples or subgraphs contain sufficient relational information to logically infer the gold answer $y$ for a given query $q$, either directly through explicit facts or indirectly through reasoning over connected triples. We therefore define a binary reward, the **Knowledge-Carrying Reward**, $R_C$, which measures the *deducibility* of gold answer in the constructed KG. An external LLM judge is

prompted with $(q, y, \mathcal{G})$ and determines whether $y$ can be deduced from the retrieved triples:

$$R_{\mathrm{C}}(q, y, \mathcal{G}) = \mathbb{I}\big[deducible(q, y \mid \mathcal{G})\big]. \tag{1}$$

***Graph-based Text Retriever*** Retrieval operates over the graph structure of the KG to locate relevant text passages. To enhance the *knowledge-indexing capability*, we use **Knoweldge-Indexing Reward**, $R_I$, as the reward function. This aligns with the fundamental objective of these retrievers by measuring the effectiveness of the retrieved passages in capturing the relevant information. The reward is defined as follows:

$$R_{\mathrm{I}}(q, \mathcal{D}_{\mathrm{gold}}, \mathcal{G}) = \frac{|\mathrm{Top}\text{-}k(\mathcal{G}, q) \cap \mathcal{D}_{\mathrm{gold}}|}{|\mathcal{D}_{\mathrm{gold}}|} \tag{2}$$

where $\mathcal{D}_{\mathrm{gold}}$ denotes the gold passages for $q$, and Top-$k(\mathcal{G}, q)$ are the retrieved passages.

## 4.3 GRPO FOR GRAPH CONSTRUCTION

To optimize the knowledge graph (KG) constructor policy $\pi_\theta^{KG}$, we employ **Group-Relative Policy Optimization (GRPO)** (Shao et al., 2024), a memory-efficient alternative to Proximal Policy Optimization (PPO). GRPO is well-suited for our LLM-based framework, as it eliminates the need for a separate value model by using a relative reward baseline derived from a group of sampled graph outputs. This approach reduces computational overhead and memory usage, enabling scalable training for large-scale graph construction tasks.

The GRPO objective updates $\pi_\theta^{KG}$ to favor graphs that maximize downstream QA performance, incorporating a clipping mechanism to ensure stable updates. We simplify the training procedure by removing the KL divergence term to lower the computational overhead and save memory usage without damaging the training (Liu et al., 2025; Hu et al., 2025). Formally, the objective is defined as:

$$\mathcal{J}_{\mathrm{GRPO}}(\theta) = \mathbb{E}_{\mathbf{s}\sim\mathcal{D}, \{\mathbf{a}_i\}_{i=1}^G \sim \pi_{\theta_{\mathrm{old}}}(\cdot|\mathbf{s})} \left[ \frac{1}{G} \sum_{i=1}^G \sum_{t=1}^{|\mathbf{a}_i|} \min\left( r_{i,t}(\theta)\hat{A}_i, \mathrm{clip}(r_{i,t}(\theta), 1-\epsilon, 1+\epsilon)\hat{A}_i \right) \right]$$

Here, the policy $\pi_\theta^{KG}$ generates a graph $\mathcal{G}$ token by token, where $\mathbf{a} = \{a_1, ..., a_T\}$ represents the sequence of tokens forming the graph. In this equation, $\mathbf{s}$ is the input document list $\mathcal{D}$, and $\mathbf{a}_i$ is the $i$-th sampled graph output from a group of $G$ samples. The probability ratio $r_{i,t}(\theta)$ is defined as $\frac{\pi_\theta(\mathbf{a}_{i,t}|\mathbf{s}, \mathbf{a}_{i,<t})}{\pi_{\theta_{\mathrm{old}}}(\mathbf{a}_{i,t}|\mathbf{s}, \mathbf{a}_{i,<t})}$. $\hat{A}_i = \frac{R_i - \mu_R}{\sigma_R}$ represents the Group-Relative Advantage for the entire graph $\mathbf{a}_i$, which is derived by normalizing its reward relative to the group's mean $\mu_R$ and standard deviation $\sigma_R$. The downstream reward signal, $R_i$, for the i-th graph sample is determined by the specific training setup: $R = R_{\mathrm{C}}$ (Eq. 1) when employing a graph knowledge retriever, or $R = R_{\mathrm{I}}$ (Eq. 2) when a graph-based text retriever is used. $\epsilon$ is a small clipping hyperparameter that ensures stable updates by preventing the new policy from straying too far from the old policy.

## 5 EXPERIMENTS

Our experiments are designed to answer three primary research questions: **RQ1.** Does optimizing KG construction with a downstream task reward (**AutoGraph-R1**) lead to better end-to-end RAG performance compared to standard, task-agnostic KG construction? **RQ2.** Is this performance improvement consistent across different graph-based RAG paradigms (i.e., when the graph is a *knowledge carrier* vs. a *knowledge index*) and across different model scales? **RQ3.** Does optimizing for downstream utility also improve the *intrinsic quality* (e.g., factual precision and recall) of the graph, and how do different reward functions bias the final graph structure?

### 5.1 DATASETS AND CORPORA

**Training Datasets** For the reinforcement learning phase, we utilize two multi-hop QA datasets: HotpotQA (Yang et al., 2018) and Musique (Trivedi et al., 2022). To create a more challenging training environment for the text retrieval scenario, we implement a hard negative mining strategy.

For each query, we use a `Qwen3-8B` (Zhang et al., 2025c) embedding model to identify the most semantically similar non-gold passage from the corpus, which is then added as a distractor. In contrast, for the graph knowledge retriever scenario, no distractors are used, as the primary objective is to optimize the informational completeness of the graph constructed from source documents, improving the knowledge-carrying capability of graph, rather than its ability to filter irrelevant content.

**Evaluation Datasets** For final RAG evaluation, we use a diverse set of five QA datasets, each comprising 1,000 samples. These include two general QA benchmarks, Natural Questions (NQ) (Kwiatkowski et al., 2019) and PopQA (Mallen et al., 2023), and three multi-hop QA benchmarks, HotpotQA (Yang et al., 2018), 2WikiMultihopQA (Ho et al., 2020), and Musique (Trivedi et al., 2022). These datasets enable a comprehensive evaluation of AutoGraph-R1 on downstream task-utility.

**Corpora.** For the general QA datasets (NQ, PopQA), the knowledge corpus is built from the introductory sections of the December 2021 Wikipedia dump Izacard et al. (2022). For the multi-hop QA datasets, the corpus for each is constructed from the documents associated with its 1,000 evaluation samples, following the methodology in Gutiérrez et al. (2025a).

## 5.2 EXPERIMENTS CONFIGS

**Models** We experiment with fine-tuning both `Qwen2.5-3B` and `Qwen2.5-7B` (Qwen et al., 2025) as the KG construction model ($\pi_\theta^{KG}$). For all RAG evaluations, we use a frozen `Qwen2.5-7B` as the answer generation LLM. The `Qwen3-0.6B` model is used consistently for all embedding tasks in both training and evaluation.

**RL Training Configuration** We fine-tune the KG construction policy using the GRPO algorithm (Shao et al., 2024) on two `H100` GPUs. For each training sample, the policy iteratively generates triples for each document. The training setup is tailored to the retrieval paradigm. For the graph-based text retriever, the model operates on a fixed pool of 15 documents per query, retrieving the top-$N$ passages, where $N$ equals the number of gold supporting passages. For the graph knowledge retriever, we retrieve an $n$-hop subgraph, where $n$ is dynamically set to match the number of hops required by the query to ensure sufficient context.

**Evaluation Protocol** For evaluation, a KG is first constructed over the entire document corpus for a each dataset. Then, depends on the type of retriever, the corresponding RAG is performed using this static graph. We report the final answer F1 score as the primary metric, consistent with prior work.

## 5.3 BASELINES

We evaluate AutoGraph-R1 by using a suite of state-of-the-art retrieval methods to benchmark the functional utility of the knowledge graphs it generates against a task-agnostic baseline.

**KG Construction Baseline** To benchmark the performance of our RL-optimized constructor, we establish a baseline using a zero-shot approach. Specifically, we construct the baseline knowledge graphs using `Qwen2.5-3B` and `Qwen2.5-7B` model guided solely by the same designed prompts used in fine-tuning models. This represents a standard, task-agnostic method for KG construction and allows us to directly measure the gains attributable to our downstream-aware optimization.

**RAG Method Baselines** We evaluate the KGs constructed by both AutoGraph-R1 and the zero-shot baseline using a suite of state-of-the-art RAG methods to measure their functional utility. For **graph knowledge retrieval**, where the graph itself is the source of information, we test three distinct approaches. First, we use **ToG** (Sun et al., 2024), setting both the width and depth to 3. It performs beam search on the graph and uses the `Qwen3-0.6B` model for relation pruning to discover meaningful paths. Second, we employ a **Subgraph Retriever**, which first performs Named Entity Recognition (NER) on the query and then expands 1-hop from the identified entities to form the retrieval context. Third, we include a **Dense Triple Retriever**, which uses the `Qwen3-0.6B` model to retrieve triples based on the semantic similarity between their embeddings and the query embedding. For these methods, the top-10 retrieved paths or triples are used as context. For **graph-based text retrieval**, where the graph serves as an index over a text corpus, we use **HippoRAG** (Gutiérrez et al., 2025a) and **HippoRAG-2** (Gutiérrez et al., 2025b). Both methods perform query-to-edge retrieval to identify relevant triples, which then seed a Personalized PageRank (PPR) algorithm over the KG to score and rank text passages. The top-5 ranked passages are then returned as evidence.

Table 1: Impact of our Knowledge-Carrying reward finetuned KG constructor on Graph Knowledge Retriever performance. The table reports the final RAG F1 scores on five QA datasets. We compare KGs generated by our fine-tuned models ("Ours") with KGs from the base models ("Base"). Fine-tuning the KG constructor with GRPO yields substantial improvements for all tested graph retrieval methods and model sizes.

| Methods | Simple QA | | Multihop QA | | | Overall |
|---|---|---|---|---|---|---|
| | NQ* | PopQA* | HotpotQA | 2WikiQA | Musique | Avg. |
| *Qwen2.5-3B* | | | | | | |
| Subgraph (Base) | 26.43 | 54.48 | 39.02 | 34.15 | 13.82 | 33.58 |
| Subgraph (Ours) | **28.03**$\uparrow_{1.60}$ | **59.46**$\uparrow_{4.98}$ | **40.77**$\uparrow_{1.75}$ | **34.71**$\uparrow_{0.56}$ | **15.13**$\uparrow_{1.31}$ | **35.62**$\uparrow_{2.04}$ |
| Triples Retriever (Base) | 30.53 | 51.67 | 40.76 | 32.18 | 17.81 | 34.58 |
| Triples Retriever (Ours) | **33.67**$\uparrow_{3.14}$ | **56.76**$\uparrow_{5.09}$ | **46.94**$\uparrow_{6.18}$ | **36.09**$\uparrow_{3.91}$ | **21.41**$\uparrow_{3.60}$ | **38.97**$\uparrow_{4.39}$ |
| ToG (Base) | 26.32 | 54.92 | 41.77 | 43.54 | 18.21 | 36.95 |
| ToG (Ours) | **29.27**$\uparrow_{2.95}$ | **61.40**$\uparrow_{6.48}$ | **44.56**$\uparrow_{2.79}$ | **49.33**$\uparrow_{5.79}$ | **18.42**$\uparrow_{0.21}$ | **40.60**$\uparrow_{3.65}$ |
| *Qwen2.5-7B* | | | | | | |
| Subgraph (Base) | 28.07 | 55.43 | 41.66 | 33.97 | 15.24 | 34.87 |
| Subgraph (Ours) | **28.54**$\uparrow_{0.47}$ | **60.94**$\uparrow_{5.51}$ | **43.59**$\uparrow_{1.93}$ | **37.43**$\uparrow_{3.46}$ | **15.65**$\uparrow_{0.41}$ | **37.23**$\uparrow_{2.36}$ |
| Triples Retriever (Base) | 33.26 | 55.56 | 44.99 | 35.57 | 20.43 | 37.96 |
| Triples Retriever (Ours) | **33.98**$\uparrow_{0.72}$ | **58.02**$\uparrow_{2.46}$ | **48.28**$\uparrow_{3.29}$ | **36.04**$\uparrow_{0.47}$ | **20.56**$\uparrow_{0.13}$ | **39.38**$\uparrow_{1.42}$ |
| ToG (Base) | 25.59 | 57.53 | 43.93 | 46.03 | 18.46 | 38.31 |
| ToG (Ours) | **29.36**$\uparrow_{3.77}$ | **62.85**$\uparrow_{5.32}$ | **44.68**$\uparrow_{0.75}$ | **50.20**$\uparrow_{4.17}$ | **19.31**$\uparrow_{0.85}$ | **41.28**$\uparrow_{2.97}$ |

Table 2: Impact of our Knowledge-Indexing reward finetuned KG constructor on Graph-based Text Retriever performance. The table shows the final RAG F1 scores, where the KG serves as an index to retrieve text passages. KGs built with our fine-tuned models ("Ours") lead to superior retrieval accuracy compared to KGs from base models ("Base"), improving results for both HippoRAG and HippoRAG2.

| Methods | Simple QA | | Multihop QA | | | Overall |
|---|---|---|---|---|---|---|
| | NQ* | PopQA* | HotpotQA | 2WikiQA | Musique | Avg. |
| *Qwen2.5-3B* | | | | | | |
| HippoRAG (Base) | 36.28 | 65.55 | 53.22 | 48.97 | 27.44 | 46.29 |
| HippoRAG (Ours) | **38.28**$\uparrow_{2.00}$ | **65.93**$\uparrow_{0.38}$ | **55.39**$\uparrow_{2.17}$ | **51.69**$\uparrow_{2.72}$ | **28.11**$\uparrow_{0.67}$ | **47.88**$\uparrow_{1.59}$ |
| HippoRAG2 (Base) | 35.88 | 65.02 | 53.70 | 50.98 | 25.70 | 46.25 |
| HippoRAG2 (Ours) | **38.45**$\uparrow_{2.57}$ | **66.23**$\uparrow_{1.21}$ | **56.28**$\uparrow_{2.58}$ | **52.80**$\uparrow_{1.82}$ | **27.93**$\uparrow_{2.23}$ | **48.34**$\uparrow_{2.09}$ |
| *Qwen2.5-7B* | | | | | | |
| HippoRAG (Base) | 37.16 | 65.95 | 55.50 | 53.01 | 26.03 | 47.53 |
| HippoRAG (Ours) | **38.80**$\uparrow_{1.64}$ | **67.85**$\uparrow_{1.90}$ | **57.19**$\uparrow_{1.69}$ | **53.60**$\uparrow_{0.59}$ | **26.97**$\uparrow_{0.94}$ | **48.88**$\uparrow_{1.35}$ |
| HippoRAG2 (Base) | 37.02 | 65.74 | 57.08 | 54.99 | 26.77 | 48.32 |
| HippoRAG2 (Ours) | **38.68**$\uparrow_{1.66}$ | **67.72**$\uparrow_{1.97}$ | **58.98**$\uparrow_{1.90}$ | **56.46**$\uparrow_{1.47}$ | **27.18**$\uparrow_{0.41}$ | **49.80**$\uparrow_{1.48}$ |

## 5.4 RESULTS AND ANALYSIS

***AutoGraph-R1 consistently improves downstream RAG performance across different paradigms and model scales.*** Our primary finding is that optimizing KG construction for downstream utility leads to significant end-to-end F1 score improvements over a standard zero-shot constructor. As shown in Table 1, when the KG acts as a *knowledge carrier*, our method yields average F1 gains of up to +4.39 (3B model) and +2.97 (7B model). Similarly, when the KG is a *knowledge index* (Table 2), performance increases by up to +2.09 and +1.48 average F1 points for the 3B and 7B models, respectively. This confirms that task-aware optimization is broadly effective, enhancing utility regardless of the graph's function or the base model's size.

***AutoGraph-R1 demonstrably improves the graph's core function as a knowledge index.*** The magnitude of F1 gains is more modest in the text retrieval setting. This is an expected outcome, stemming from the dual nature of using full text passages as evidence. On one hand, their rich context can enable the generator to succeed even with imperfect retrieval, masking some F1 gains. On the other hand, this verbosity can introduce noise, unlike the concise, structured triples provided by the graph knowledge retriever. To isolate the direct impact on retrieval quality, we evaluate passage

recall@5. Table 3 shows clear improvements: average recall increases by over 2 points for both 3B and 7B model. This confirms that our RL framework creates a more effective knowledge index.

Table 3: Evaluating Knowledge Indexing Quality via Passage Recall. This table demonstrates that KGs constructed by AutoGraph-R1 consistently improve passage recall@5 over zero-shot baselines when used with graph-based text retrievers (HippoRAG and HippoRAG2). This confirms the RL-optimized graph is a more effective knowledge index for guiding retrieval.

| Methods | Simple QA | | Multihop QA | | | Overall |
|---|---|---|---|---|---|---|
| | NQ* | PopQA* | HotpotQA | 2WikiQA | Musique | Avg. |
| *Qwen2.5-3B* | | | | | | |
| HippoRAG (Base) | 79.50 | 92.10 | 68.41 | 70.84 | 46.5 | 71.47 |
| HippoRAG (Ours) | **93.00**↑$_{13.5}$ | **95.60**↑$_{4.20}$ | **68.82**↑$_{0.41}$ | **74.02**↑$_{3.18}$ | **47.65**↑$_{1.15}$ | **75.82**↑$_{4.35}$ |
| HippoRAG2 (Base) | 82.20 | 92.20 | 70.06 | 73.49 | 46.93 | 72.98 |
| HippoRAG2 (Ours) | **94.00**↑$_{11.8}$ | **95.40**↑$_{3.40}$ | **71.21**↑$_{1.15}$ | **76.42**↑$_{2.93}$ | **49.13**↑$_{2.2}$ | **77.23**↑$_{4.25}$ |
| *Qwen2.5-7B* | | | | | | |
| HippoRAG (Base) | 93.30 | 92.90 | 69.97 | 72.87 | 47.19 | 75.24 |
| HippoRAG (Ours) | **94.30**↑$_{1.0}$ | **95.80**↑$_{2.9}$ | **71.4**↑$_{1.43}$ | **76.16**↑$_{3.29}$ | **48.44**↑$_{1.25}$ | **77.22**↑$_{1.98}$ |
| HippoRAG2 (Base) | 94.10 | 92.80 | 72.03 | 75.98 | 48.55 | 76.69 |
| HippoRAG2 (Ours) | **95.00**↑$_{0.9}$ | **96.30**↑$_{3.5}$ | **73.61**↑$_{1.58}$ | **78.66**↑$_{2.68}$ | **49.23**↑$_{0.68}$ | **78.56**↑$_{1.87}$ |

***Optimizing for downstream utility also enhances the intrinsic factual quality of the graph.*** We investigated whether extrinsic optimization comes at the cost of intrinsic quality by measuring the precision, recall, and F1 score of the extracted triples against the source text (Huang et al., 2025a) using Deepseek-V3 model as a judge (DeepSeek-AI et al., 2025b). The results in Table 4 show a clear positive correlation. Across all datasets, KGs fine-tuned with AutoGraph-R1 exhibit higher intrinsic F1 scores than their zero-shot counterparts. This indicates our RL framework does not sacrifice factual accuracy for functional utility; rather, it improves both simultaneously.

***The choice of reward function induces specific and beneficial structural biases in the KG.*** A deeper analysis of Table 4 reveals that the two reward functions specialize the graph's structure. The **Knowledge-Carrying Reward** ($R_C$), optimized for graph knowledge retrieval, consistently produces graphs with higher recall, aligning with its goal of ensuring all necessary facts for reasoning are present. In contrast, the **Knowledge-Indexing Reward** ($R_I$), optimized for text retrieval, yields graphs with higher precision, reflecting its need for a clean, high-fidelity index. This finding highlights that AutoGraph-R1 not only improves graph quality but also tailors the graph's structure to its specific downstream function.

Table 4: Further analysis on whether GRPO training increases the triple extraction quality, measured by Precision, Recall and, F1 defined in previous work (Huang et al., 2025a; Bai et al., 2025).

| KG Construction Model | HotpotQA | | | 2WikiMultihopQA | | | Musique | | | 2021Wiki | | |
|---|---|---|---|---|---|---|---|---|---|---|---|---|
| | Acc | Recall | F1 | Acc | Recall | F1 | Acc | Recall | F1 | Acc | Recall | F1 |
| Qwen2.5-7B-Instruct | 98.50 | 93.68 | 95.65 | 94.80 | 91.19 | 92.68 | 96.77 | 95.27 | 95.73 | 95.03 | 91.39 | 92.92 |
| + GRPO with Knowledge-Carrying Reward | 97.53 | **96.66** | **96.71** | 95.51 | **96.55** | 95.25 | 97.14 | **96.75** | **96.45** | 96.17 | **96.66** | **96.15** |
| + GRPO with Knowledge-Indexing Reward | **98.96** | 94.81 | 96.59 | **98.35** | 94.54 | **96.16** | **99.53** | 93.14 | 95.81 | **97.44** | 95.01 | 95.99 |
| Qwen2.5-3B-Instruct | 94.41 | 91.00 | 91.92 | 83.53 | 79.34 | 81.01 | 92.07 | 89.52 | 90.31 | 87.79 | 86.01 | 86.63 |
| + GRPO with Knowledge-Carrying Reward | 96.52 | **94.24** | 94.80 | 95.91 | **96.28** | **95.80** | **97.01** | **94.74** | **95.22** | 96.70 | 95.58 | 95.76 |
| + GRPO with Knowledge-Indexing Reward | **97.11** | 93.15 | **94.64** | **96.19** | 93.66 | 94.48 | 96.20 | 93.87 | 94.55 | **98.22** | **96.04** | **96.85** |

# 6 CONCLUSION

In this work, we introduced **AutoGraph-R1**, the first reinforcement learning framework for knowledge graph construction that directly optimizes downstream RAG performance. By incorporating task-aware rewards, our approach bridges the gap between traditional graph quality metrics and end-task utility. Experiments across five QA benchmarks demonstrate consistent improvements over strong baselines in both graph knowledge and graph-based text retrieval. Overall, our work shows that reinforcement learning can effectively connect the graph construction process with downstream QA performance, ensuring that knowledge graphs are optimized for their intended applications.

## 7 ETHICS STATEMENT

We affirm our commitment to the ICLR Code of Ethics. Our research does not involve human subjects or the collection of new personally identifiable information. All datasets used for training (HotpotQA, Musique) and evaluation (NQ, PopQA, etc.) are publicly available benchmarks and were used in accordance with their licenses. All models employed (Qwen series, DeepSeek-V3) are open-source and were run locally. We use an open-source RL framework VeRL (Sheng et al., 2025) for training. While our work aims to improve the factuality of LLMs, we acknowledge that the underlying models and data can contain biases, which may be reflected in the generated graphs. Experiments were conducted on two H100 GPUs; we have focused on models in the 3B-7B parameter range to promote accessible research.

## 8 REPRODUCIBILITY STATEMENT

**Models** Key models in our experiments include `Qwen2.5-3B` and `Qwen2.5-7B` as KG constructors, a frozen `Qwen2.5-7B` for answer generation, and `Qwen3` series models for embeddings. `DeepSeek-V3` was used as the LLM judge for evaluation. All models is open-source and available via the Hugging Face Hub.

**Code, Checkpoints, and Data.** All datasets are standard public benchmarks. We will release our full source code, including the custom reinforcement learning (RL) training loop, retriever implementations, evaluation scripts, and specific prompts for baselines. Crucially, we will also release the final checkpoints for our trained AutoGraph-R1 constructor models upon publication, allowing for the direct replication of our results. All hyperparameter configurations will be provided in the released code.

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

## A    TRAINING DYNAMICS OF AUTOGRAPH-R1

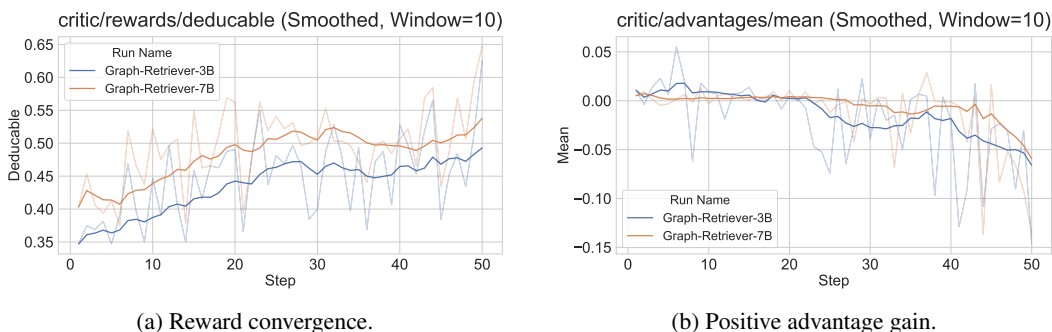

(a) Reward convergence.                                                    (b) Positive advantage gain.

Figure 3: **Effective Training Dynamics with the Deducible Reward** ($R_C$). Training curves for the Graph Knowledge Retriever setting. (a) The reward, measuring answer deducibility, steadily increases and converges, demonstrating the policy is successfully learning its objective. (b) The advantage gain trends towards a small negative value, indicating that the value function's estimate of expected reward is rising quickly while the policy makes stable, incremental improvements. This dynamic, coupled with the rising absolute reward, points to effective and controlled optimization.

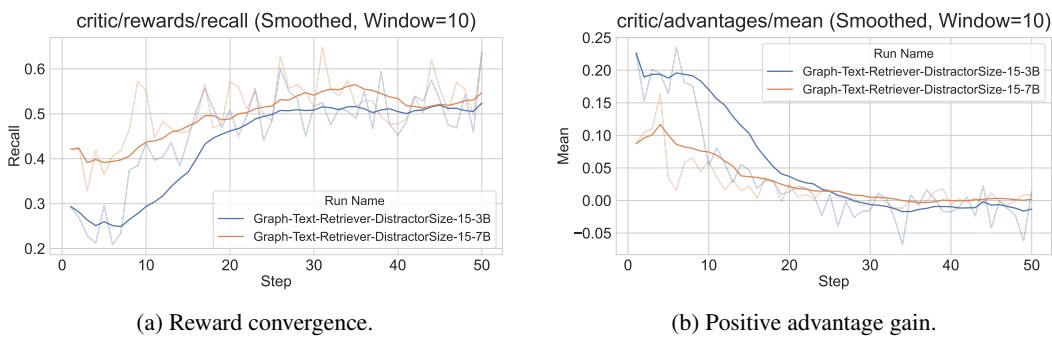

(a) Reward convergence.                                                    (b) Positive advantage gain.

Figure 4: **Effective Training Dynamics with the Knowledge-Indexing Reward** ($R_I$). Training curves for the Graph-based Text Retriever setting. (a) The reward, measuring passage recall, shows a clear upward trend of improvement. (b) The advantage gain dynamic, paired with the rising reward curve, confirms that the policy is effectively learning from this stable, task-specific signal.

## B    IMPACT OF USING F1 REWARD FOR AUTOGRAPH-R1

To validate our choice of using task-specific rewards ($R_C$ and $R_I$), we conducted an ablation study comparing them against a more direct but potentially noisier signal: the final answer's F1 score. We trained two additional KG constructor models using the RAG F1 score as the reward signal. The results demonstrate that our proposed task-specific rewards are significantly more effective and stable, a finding supported by both the training dynamics and final performance metrics.

***F1-based RL leads to unstable training and poor performance.*** Figure 5a illustrates the instability inherent in using the final F1 score as a reward. The reward curve (Figure 5a) exhibits high variance and lacks a clear, monotonic upward trend comparing with using task specific reward, indicating a noisy learning signal.

The downstream impact of this unstable training is evident in Tables 5 and 6. For graph knowledge retrievers, the F1-rewarded model yields inconsistent results and, in the case of the Triples Retriever, underperforms the zero-shot baseline by over -2.2 avg. F1 points. This contrasts sharply with our **Knowledge-Carrying Reward** ($R_C$), which delivers consistent gains across all retriever types. The failure is even more pronounced for graph-based text retrievers. The F1-rewarded model degrades average performance below the baseline for both HippoRAG and HippoRAG2. Table 7 reveals why

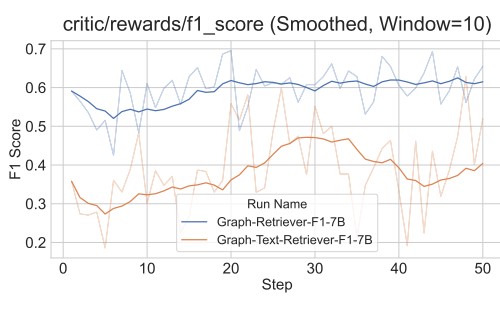 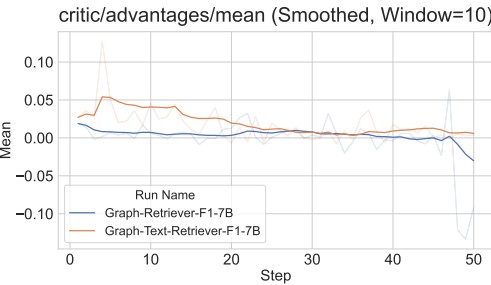

(a) Volatile reward signal.         (b) Stagnant advantage gain.

Figure 5: **Unstable Training Dynamics Using a Naive F1 Reward.** Training curves for the ablation study where the final RAG F1 score is used as the reward. (a) The F1 reward signal is highly volatile and shows no clear upward trend, providing a noisy and ineffective learning signal. (b) Consequently, the advantage gain remains flat and centered around zero, confirming that the policy is failing to find a consistent direction for improvement. This leads to stalled optimization, as reflected in the poor downstream results.

Table 5: Ablation Study on Reward Functions for **Graph Knowledge Retrievers**. This table compares the final RAG F1 scores of KGs built using a zero-shot baseline, our proposed **Deducible Reward** ($R_C$), and a naive **F1 Reward**. Results are for the Qwen2.5-7B model. Our task-specific Deducible Reward consistently outperforms the unstable F1 Reward.

| Methods | Simple QA | | Multihop QA | | | Overall |
|---|---|---|---|---|---|---|
| | NQ* | PopQA* | HotpotQA | 2WikiQA | Musique | Avg. |
| *Qwen2.5-7B* | | | | | | |
| Subgraph (Base) | 28.07 | 55.43 | 41.66 | 33.97 | 15.24 | 34.87 |
| Subgraph (F1 Reward) | 27.36 | 55.09 | 41.15 | 35.13 | 15.70 | 34.89 |
| Subgraph (Knowledge-Carrying Reward) | **28.54** | **60.94** | **43.59** | **37.43** | **15.65** | **37.23** |
| Triples Retriever (Base) | 33.26 | 55.56 | 44.99 | 35.57 | 20.43 | 37.96 |
| Triples Retriever (F1 Reward) | 31.52 | 53.85 | 44.52 | 30.68 | 18.00 | 35.71 |
| Triples Retriever (Knowledge-Carrying Reward) | **33.98** | **58.02** | **48.28** | **36.04** | **20.56** | **39.38** |
| ToG (Base) | 25.59 | 57.53 | 43.93 | 46.03 | 18.46 | 38.31 |
| ToG (F1 Reward) | 27.64 | 56.95 | 45.19 | 51.10 | 18.37 | 39.85 |
| ToG (Knowledge-Carrying Reward) | **29.36** | **62.85** | **44.68** | **50.20** | **19.31** | **41.28** |

Table 6: Ablation Study on Reward Functions for **Graph-based Text Retrievers**. This table compares the final RAG F1 scores using KGs from a zero-shot baseline, our **Recall Reward** ($R_I$), and a naive **F1 Reward**. Results are for the Qwen2.5-7B model. The F1 Reward signal is unstable and degrades performance, while our task-specific Recall Reward provides consistent gains.

| Methods | Simple QA | | Multihop QA | | | Overall |
|---|---|---|---|---|---|---|
| | NQ* | PopQA* | HotpotQA | 2WikiQA | Musique | Avg. |
| *Qwen2.5-7B* | | | | | | |
| HippoRAG (Base) | 37.16 | 65.95 | 55.50 | 53.01 | 26.03 | 47.53 |
| HippoRAG (F1 Reward) | **39.66** | 63.74 | 53.74 | 49.58 | **28.68** | 47.08 |
| HippoRAG (Knowledge-Indexing Reward) | 38.80 | **67.85** | **57.19** | **53.60** | 26.97 | **48.88** |
| HippoRAG2 (Base) | 37.02 | 65.74 | 57.08 | 54.99 | 26.77 | 48.32 |
| HippoRAG2 (F1 Reward) | 38.33 | 62.92 | 55.19 | 50.23 | **27.51** | 46.83 |
| HippoRAG2 (Knowledge-Indexing Reward) | **38.68** | **67.72** | **58.98** | **56.46** | 27.18 | **49.80** |

using the F1 reward actively hurts retrieval quality, causing a drop in average recall@5. This suggests the optimizer, chasing a volatile signal, creates a graph that is structurally worse for retrieval.

In contrast, our proposed **Knowledge-Indexing Reward, ($R_I$)** provides consistent and positive gains in both final F1 score and, critically, in the underlying recall@5 metric. Overall, this two-part analysis provides compelling evidence that a direct, task-specific reward measuring a graph's functional utility

Table 7: Ablation Study on Reward Functions for **Knowledge Indexing Quality (Recall@5)**. This table shows the direct impact of the reward function on the graph's ability to guide text retrieval. While our task-specific **Recall Reward** ($R_I$) consistently improves recall, the naive **F1 Reward** often degrades it below the baseline, highlighting its unsuitability for this task.

| Methods | Simple QA | | Multihop QA | | | Overall |
|---|---|---|---|---|---|---|
| | NQ* | PopQA* | HotpotQA | 2WikiQA | Musique | Avg. |
| *Qwen2.5-7B* | | | | | | |
| *HippoRAG* | | | | | | |
| HippoRAG (Base) | 93.30 | 92.90 | 69.97 | 72.87 | 47.19 | 75.25 |
| HippoRAG (F1 Reward) | 93.90 | 92.10 | 69.18 | 72.71 | 47.71 | 75.12 |
| HippoRAG (Recall Reward) | **94.30** | **95.80** | **71.40** | **76.16** | **48.44** | **77.22** |
| *HippoRAG2* | | | | | | |
| HippoRAG2 (Base) | 94.10 | 92.80 | 72.03 | 75.98 | 48.55 | 76.69 |
| HippoRAG2 (F1 Reward) | 94.80 | 92.00 | 71.07 | 72.81 | 48.75 | 75.88 |
| HippoRAG2 (Recall Reward) | **95.00** | **96.30** | **73.61** | **78.66** | **49.23** | **78.56** |

is a more stable and effective signal for RL-based KG construction than a sparse and noisy end-to-end task metric.

## C  CASE STUDIES: THE FUNCTIONAL ADVANTAGE OF AUTOGRAPH-R1

To qualitatively illustrate the benefits of our task-aware optimization, we present two case studies from the 2WikiMultiHopQA dataset that highlight how AutoGraph-R1 constructs functionally superior knowledge graphs compared to a standard zero-shot baseline.

### C.1  CASE STUDY 1: COMPARATIVE REASONING

The first case study examines a question requiring a comparison between the death dates of two film directors. This task requires the KG to contain specific, comparable facts (i.e., dates) for multiple entities. As shown in Figure 6, the zero-shot KG fails because it does not extract the specific death dates needed for comparison. In contrast, the KG constructed by AutoGraph-R1 contains the necessary date information, as the RL training has taught the constructor that dates are critical for such questions. This complete evidence enables the LLM to easily answer the question correctly.

### C.2  CASE STUDY 2: PATH-BASED REASONING

The second case study involves a 2-hop question that requires finding a path from a film to its director, and then from the director to their child. This task depends on the structural connectivity of the graph.

As shown in Figure 7, the zero-shot KG (top) fails critically. While it successfully extracts the first link in the path—'(Los Pagares de Mendieta, directed by, Leopoldo Torres Ríos)'—it fails to extract the second, crucial link about the director's child. The reasoning path is broken after the first hop, causing the QA system to fail. In contrast, the AutoGraph-R1 KG (bottom) explicitly contains the complete 2-hop reasoning path. It successfully extracts both '(Los Pagares de Mendieta, directed by, Leopoldo Torres Ríos)' and '(Leopoldo Torres Ríos, father of, Leopoldo Torre Nilsson)'. The RL process has rewarded the constructor for building these essential connective trails, recognizing that entity linkage across different relationships is crucial for multi-hop QA.

## D  AUTOGRAPH-R1 TRAINING ALGORITHM

The end-to-end training process for the AutoGraph-R1 KG constructor is formalized in Algorithm 1. The core idea is to iteratively construct a knowledge graph for a given query and its context documents, evaluate the graph's utility using a task-specific reward function, and then update the constructor's policy using the collected rewards.

---

**Case Study 1: Zero-Shot KG (ToG Retriever) - Failed Answer**

**Question:** Which film has the director who died first, The Goose Woman or You Can No Longer Remain Silent?

**Retrieved Triples:**
```
"(You Can No Longer Remain Silent, directed by, Robert A. Stemmle)",
"(Robert A. Stemmle, died in, Baden-Baden, Germany)",
"(The Goose Woman, directed by, Clarence Brown)",
"(Clarence Brown, was a, American film director)"
... (and other irrelevant triples)
```

---

**Case Study 1: AutoGraph-R1 KG (ToG Retriever) - Correct Answer**

**Question:** Which film has the director who died first, The Goose Woman or You Can No Longer Remain Silent?

**Retrieved Triples:**
```
"(You Can No Longer Remain Silent, directed by, Robert A. Stemmle)",
"(Robert A. Stemmle, died on, 24 February 1974)",
"(The Goose Woman, directed by, Clarence Brown)",
"(Clarence Brown, died on, August 17, 1987)",
...
```

---

Figure 6: Qualitative comparison for a **comparative reasoning** question. The zero-shot KG lacks specific death dates, leading to failure. The AutoGraph-R1 KG, optimized for task utility, successfully extracts the critical dates needed for comparison.

---

**Case Study 2: Zero-Shot KG (ToG Retriever) - Failed Answer**

**Question:** Who is the child of the director of film Los Pagares De Mendieta?

**Retrieved Triples:**
```
"(Los Pagares de Mendieta, directed by, Leopoldo Torres RŎ0edos)",
"(Leopoldo Torres RŎ0edos, age at death, 60)",
"(Leopoldo Torres RŎ0edos, occupation, film director and
screenwriter)",
... (and other facts about the director, but not their child)
```

---

**Case Study 2: AutoGraph-R1 KG (ToG Retriever) - Correct Answer**

**Question:** Who is the child of the director of film Los Pagares De Mendieta?

**Retrieved Triples:**
```
"(Los Pagares de Mendieta, directed by, Leopoldo Torres RŎ0edos)",
"(Leopoldo Torres RŎ0edos, father of, Leopoldo Torre Nilsson, ...)",
"(Leopoldo Torres RŎ0edos, born on, 27 December 1899)",
...
```

---

Figure 7: Qualitative comparison for a **2-hop path-based** question. The zero-shot KG extracts the first link (director of the film) but misses the second (child of the director), breaking the reasoning path. The AutoGraph-R1 KG successfully constructs the full path.

# E  PROMPTS

This section details the specific prompts used in our experimental pipeline. The process begins with the graph construction prompt (Figure 8), which guides the LLM to extract triples from raw text. During RL training, the **Knowledge-Carrying Reward** ($R_C$) is determined using the deducibility judge prompt shown in Figure 9. For the final RAG answer generation step, we use distinct prompts tailored to the retrieved context: one for linearized graph triples (Figure 10) and another for raw text passages (Figure 11). Finally, Figure 12 shows the prompts used for our intrinsic graph quality analysis, where an LLM judge generates and answers multiple-choice questions to evaluate factual coverage.

---

**Algorithm 1** AutoGraph-R1 Training Loop

---

1: **Input:** Training dataset $\mathcal{S} = \{(q_i, y_i, \mathcal{D}_{q_i})\}_{i=1}^{N}$, where $\mathcal{D}_{q_i}$ are the context documents for query $q_i$.
2: **Input:** KG constructor policy $\pi_\theta^{KG}$ (an LLM).
3: **Input:** Frozen retriever $\mathcal{R}_{\text{frozen}}$ (either a graph knowledge retriever or a graph-based text retriever).
4: **Input:** Chosen reward function $R_{\text{task}}$ (either $R_C$ or $R_I$).
5: **Initialize:** Policy parameters $\theta$.
6: **for** each training step **do**
7:     Sample a minibatch of data $\{(q, y, \mathcal{D}_q)\}$ from $\mathcal{S}$.
8:     Initialize an empty list of trajectories 'trajectories'.
9:     **for** each sample $(q, y, \mathcal{D}_q)$ in the minibatch **do**
10:                                         ▷ **Step 1: Construct the Knowledge Graph**
11:         Generate the graph by sampling from the policy: $\mathcal{G} \sim \pi_\theta^{KG}(\cdot \mid \mathcal{D}_q)$.
12:                                   ▷ **Step 2: Determine Task-Specific Reward**
13:         **if** $R_{\text{task}}$ is Knowledge-Carrying Reward ($R_C$) **then**
14:             Use the frozen retriever $\mathcal{R}_{\text{graph}}$ to get evidence $\mathcal{P}(q)$ from $\mathcal{G}$.
15:             Calculate reward $r = R_C(q, y, \mathcal{P}(q))$ using Eq. (1).
16:         **else if** $R_{\text{task}}$ is Knowledge-Indexing Reward ($R_I$) **then**
17:             Use the frozen retriever $\mathcal{R}_{\text{text}}$ to get passages $\mathcal{T}(q)$ from $\mathcal{G}$.
18:             Calculate reward $r = R_I(q, y, \mathcal{T}(q))$ using Eq. (2).
19:         **end if**
20:         Store the generation trajectory (actions taken to build $\mathcal{G}$) and the final reward $r$ in 'trajectories'.
21:     **end for**
22:                                        ▷ **Step 3: Update Policy Parameters**
23:     Compute the policy gradient $\nabla_\theta J(\theta)$ using the stored 'trajectories' and a policy optimization algorithm (e.g., GRPO).
24:     Update the policy parameters: $\theta \leftarrow \theta - \eta \cdot \nabla_\theta J(\theta)$.
25: **end for**
26: **Return:** Optimized KG constructor parameters $\theta$.

---

> **Graph Construction**
>
> **Graph Generation System Prompt:**
> You are an expert knowledge graph constructor. Your task is to extract factual information from the provided text and represent it strictly as a JSON array of knowledge graph triples.
> Output Format
>   - The output must be a \*\*JSON array\*\*.
>   - Each element in the array must be a \*\*JSON object\*\* with exactly three non-empty keys:
>     - "subject": the main entity, concept, event, or attribute.
>     - "relation": a concise, descriptive phrase or verb that describes the relationship (e.g., "founded by", "started on", "is a", "has circulation of").
>     - "object": the entity, concept, value, event, or attribute that the subject has a relationship with.
> Constraints
>   - \*\*Do not include any text other than the JSON output.\*\*
>   - Do not add explanations, comments, or formatting outside of the JSON array.
>   - Extract \*\*all possible and relevant triples\*\*.
>   - All keys must exist and all values must be non-empty strings.
>   - The "subject" and "object" can be specific entities (e.g., "Radio City", "Football in Albania", "Echosmith") or specific values (e.g., "3 July 2001", "1,310,696").
>   - If no triples can be extracted, return exactly: '[]'.
> Extracts for: {passage}

Figure 8: The prompt used for both zero-shot KG construction and fine-tuning KG constructor model during RL.

---

**Deducible Judge**

**Deducible Judge Prompt:**
As an advanced reading comprehension assistant, your task is to evaluate whether the provided knowledge graph (KG) context contains sufficient information to deduce the given true answer to the question. Analyze the KG context carefully and determine if it fully supports the true answer without requiring external knowledge. Respond with only 'Yes' or 'No', indicating whether the true answer can be deduced from the KG context.
Knowledge graph (KG) context:{triples string}
Question:{query}
True Answer:{answer}
Can the true answer be deduced from the KG context? Answer 'Yes' or 'No' only.

---

Figure 9: The prompts for freeze LLM to determine the **Knowledge-Carrying Reward** ($R_C$). The 'Yes' or 'No' response serves as the binary reward signal.

---

**Graph Retriever Answer Generation**

**Answer Generation Prompt For Graph Retriever:**
As an advanced reading comprehension assistant, your task is to analyze extracted information and corresponding questions meticulously. If the knowledge graph information is not enough, you can use your own knowledge to answer the question. Your response start after "Thought: ", where you will methodically break down the reasoning process, illustrating how you arrive at conclusions. Conclude with "Answer: " to present a concise, definitive response as a noun phrase, no elaborations.
{triples string}
{question}
Thought:

---

Figure 10: The prompt used by the final answer generator when the retrieved evidence consists of linearized knowledge graph triples.

---

**Graph Text Retriever Answer Generation**

**Answer Generation Prompt:**
As an advanced reading comprehension assistant, your task is to analyze text passages and corresponding questions meticulously. If the information is not enough, you can use your own knowledge to answer the question. Your response start after "Thought: ", where you will methodically break down the reasoning process, illustrating how you arrive at conclusions. Conclude with "Answer: " to present a concise, definitive response as a noun phrase, no elaborations.
{Retrieved Texts}
{question}
Thought:

---

Figure 11: The prompt used by the final answer generator when the retrieved evidence consists of raw text passages.

---

**Multiple-Choice Question Generation and Answering**

**MCQ Generation Prompt:**
You are an expert in generating multiple-choice questions (MCQs) from scientific texts. Your task is to generate 5 multiple-choice questions based on the following passage.
Each question should:
   - Focus on factual claims, numerical data, definitions, or relational knowledge from the passage.
   - Have 4 options (one correct answer and three plausible distractors).
   - Clearly indicate the correct answer.
The output should be in JSON format, with each question as a dictionary containing:
   - "question": The MCQ question.
   - "options": A list of 4 options (e.g., ["A: ..", "B: ..", "C: ..", "D: .."]).
   - "answer": The correct answer (e.g., "A").
Passage: {passage}

---

**MCQ Answering Prompt:**
Given the contexts or evidences: {contexts}
Here is a multiple-choice question:
Question: {question}
Options: A. {options_0} B. {options_1} C. {options_2} D. {options_3}
Please select the correct answer by choosing A, B, C, or D. Respond with only the letter of your choice.

Figure 12: The prompt provided to the LLM judge (`DeepSeek-V3`) to evaluate triples extraction quality

