# OpenReview forum: "AutoGraph-R1: End-to-End Reinforcement Learning for Knowledge Graph Construction"
_ICLR.cc/2026/Conference — ICLR 2026 Conference Withdrawn Submission_

### Official Review · Reviewer_qEpk · 2025-10-24

**Soundness:** 2
**Presentation:** 3
**Contribution:** 2
**Rating:** 2
**Confidence:** 4

**Summary:**

This paper aims to build a knowledge graph (KG) for Retrieval-Augmented Generation (RAG) in an end-to-end manner. The authors propose training the KG constructor using reinforcement learning within a graph-based RAG system. They design two rewards corresponding to two applications: a graph knowledge retriever and a graph-based text retriever. The proposed method is trained on two multi-hop question answering (QA) datasets and evaluated on five QA datasets.

**Strengths:**

1. The paper is well-written.
2. The proposed approach consistently outperforms baselines.

**Weaknesses:**

1. The novelty is limited, as the paper applies standard reinforcement learning techniques to KG construction without significant methodological innovation.
2. The motivation is not clearly justified by the experiments, i.e., why is an RL-based approach necessary?

**Questions:**

1. In Section 4.2, the graph knowledge retriever reward requires computing the entire KG. Is this computationally expensive?
2. In Section 4.3, what are the input and output of the KG construction policy model $\pi_{\theta}^{KG}$?
3. The fine-tuned model is expected to outperform the zero-shot approach. The author should also include a strong pre-trained LLM as a baseline for knowledge graph construction.
4. To demonstrate the contribution of RL-based graph construction, the authors should compare against other KG construction baselines, such as HippoRAG-2 (Gutierrez et al., 2025b).
5. In Section 5.1, are there any experiments showing the effect of removing distractors during training? Also, why is this discussed in the dataset section rather than the methodology section?
6. Moreover, additional baselines are needed to justify KG construction as an intermediate step. For example, why not fine-tune the dense triple retriever directly?

---

> ### Author Response · Authors · 2025-11-23
> **Response to Weaknesses and Questions 1-2**
>
> We thank reviewer qEpK for your time and feedback. We appreciate your acknowledgment that:
>
> *   The paper is well-written.
> *   Our proposed approach consistently outperforms the baselines.
>
> We understand the reviewer's concerns regarding novelty and justification. We believe our detailed responses below, which include new supporting experiments and comparisons against state-of-the-art models, will comprehensively address these points and clearly demonstrate the novelty and necessity of our approach.
>
> We believe the following paragraphs could address the weaknesses and questions mentioned by the reviewer.
> ### **Weakness 1: Limited Novelty**
>
> > The novelty is limited, as the paper applies standard reinforcement learning techniques to KG construction without significant methodological innovation.
>
> Thank you for this point. We believe the novelty of our work lies not in inventing a new RL algorithm, but in being the **first to apply a reward-driven framework to offline KG construction for GraphRAG.**
>
> Our primary contribution and novelty are in addressing the **offline graph construction stage**. While recent works like Graph-R1 and Search-R1 apply RL to optimize *online retrieval* (e.g., query rewriting, reasoning on retrieved context), they still depend on a pre-built knowledge base. We try to tackle the foundational problem for GraphRAG: how to generate a better graph from raw documents in the first place.
>
> Our work is therefore complementary to online (e.g. Graph-R1) methods. By creating a higher-quality, task-optimized graph during the offline phase, our approach provides a stronger foundation that any subsequent online retrieval method can leverage for better performance.
>
> ### **Weakness 2: Justification for RL**
>
> > The motivation is not clearly justified by the experiments, i.e., why is an RL-based approach necessary?
>
> This is a key question that gets to the heart of our design. An RL-based approach is necessary due to the **decoupled nature of the GraphRAG pipeline.**
>
> The LLM that builds the graph (offline) has no direct connection to the retriever that uses it later (online). This means there is no gradient signal that can flow from the downstream task's success or failure back to the graph constructor. The model essentially works "in the dark," with no feedback on whether the graph it built was actually useful.
>
> Reinforcement learning is the natural and necessary solution to bridge this gap. The reward function acts as a "proxy gradient," providing a crucial feedback signal from the downstream task. This signal teaches the graph constructor what a "good" graph looks like from the retriever's perspective, allowing it to optimize its structure for final task performance.
>
> ### Questions
>
> > In Section 4.2, is computing the graph knowledge retriever reward computationally expensive?
>
> No, the computation is not a significant bottleneck in our framework. The reward calculation for one retrieval step is efficient and consists of the following steps:
> 1.  **Indexing:** We perform a fast graph indexing operation on the generated graph (typically ~50 edges) using the highly optimized Faiss library.
> 2.  **LLM Call 1:** A frozen LLM generates initial starting nodes for graph exploration.
> 3.  **LLM Call 2:** The retrieved subgraph is judged for correctness by another frozen LLM to compute the final deducible reward.
>
> Because the models are frozen and the graph is small per sample, these operations, managed asynchronously, do not introduce major latency.
>
> ---
>
> > In Section 4.3, what are the input and output of the KG construction policy model?
>
> The policy model functions as a text-to-graph generator.
> *   The **input** is a set of documents relevant to a given query.
> *   The **output** is a list of knowledge triples `(head, relation, tail)` extracted from those documents.
>
> The number of input documents varies by the training setup:
> *   **Graph Retriever setup:** The input is the set of 2-4 ground-truth supporting documents.
> *   **Graph-Text Retriever setup:** The input is a larger batch of 15 documents, comprising both supporting documents and their hard-negative distractors.

---

> ### Author Response · Authors · 2025-11-23
> **Continue: Response to Questions 3-4**
>
> > The fine-tuned model should outperform zero-shot. A strong pre-trained LLM should be included as a baseline.
>
>  We agree completely. A comparison with strong, pre-trained LLMs is essential to contextualize our results. We benchmarked our fine-tuned models against a suite of state-of-the-art large models and found that our method provides a highly parameter-efficient path to top-tier performance.
>
> **Table 1: F1 Score Comparison vs. Large Models (Graph Retrievers)**
>
> | Model | Avg F1 (Subgraph) | Avg F1 (Triples) | Avg F1 (ToG) |
> | :--- | :---: | :---: | :---: |
> | **_State-of-the-Art Zero-Shot Models_** | | | |
> | Llama-3.3-70B | 36.29 | 36.60 | 40.65 |
> | Qwen2.5-72B | 36.79 | 37.11 | 41.41 |
> | Qwen3-235B | 36.29 | 36.89 | **41.60** |
> | Deepseek-V3 | 35.85 | 37.03 | 41.20 |
> | gpt-oss-120b | 34.62 | 35.81 | 39.41 |
> | GPT-5-mini | 34.82 | 36.13 | 40.58 |
> | gemini-2.5-flash-lite | 35.91 | 37.04 | 40.06 |
> | **_Our Fine-tuned Models (GRPO)_** | | | |
> | **Qwen-3B (Ours)** | 35.62 | 38.97 | 40.60 |
> | **Qwen-7B (Ours)** | **37.23** | **39.38** | 41.28 |
>
> **Table 2: F1 Score Comparison vs. Large Models (Text Retrievers)**
>
> | Model | Avg F1 (HippoRAG) | Avg F1 (HippoRAG2) |
> | :--- | :---: | :---: |
> | **_State-of-the-Art Zero-Shot Models_** | | |
> | Llama-3.3-70B | 48.00 | 48.33 |
> | Qwen2.5-72B | 47.20 | 48.45 |
> | Qwen3-235B | 47.92 | 48.48 |
> | Deepseek-V3 | 47.56 | 47.89 |
> | gpt-oss-120b | 46.12 | 47.06 |
> | GPT-5-mini | 47.18 | 47.75 |
> | gemini-2.5-flash-lite | 47.77 | 47.97 |
> | **_Our Fine-tuned Models (GRPO)_** | | |
> | **Qwen-3B (Ours)** | 47.88 | 48.34 |
> | **Qwen-7B (Ours)** | **48.88** | **49.80** |
>
> **Table 3: Recall@5 Score Comparison vs. Large Models (Text Retrievers)**
>
> | Model | Avg Recall (HippoRAG) | Avg Recall (HippoRAG2) |
> | :--- | :---: | :---: |
> | **_State-of-the-Art Zero-Shot Models_** | | |
> | Llama-3.3-70B | 73.77 | 76.97 |
> | Qwen2.5-72B | 75.30 | 76.95 |
> | Qwen3-235B | 75.51 | 76.18 |
> | Deepseek-V3 | 75.40 | 76.48 |
> | gpt-oss-120b | 73.24 | 73.93 |
> | GPT-5-mini | 74.96 | 75.29 |
> | gemini-2.5-flash-lite | 74.85 | 76.10 |
> | **_Our Fine-tuned Models (GRPO)_** | | |
> | **Qwen-7B (Ours)** | **77.22** | **78.56** |
>
> > To demonstrate the contribution, the authors should compare against other KG construction baselines, such as HippoRAG-2.
>
> Thank you for this excellent suggestion. To provide a direct comparison, we conducted an experiment using the official HippoRAG [codebase](https://github.com/OSU-NLP-Group/HippoRAG). We replaced their default zero-shot KG constructor (Qwen2.5-7B) with our fine-tuned 7B model and measured the impact on the final retriever performance.
>
> The results below show that our fine-tuned model serves as a superior drop-in replacement, improving both F1 and Recall scores, even when using a simple prompt without the sophisticated processing of the original HippoRAG paper.
>
> **Baseline HippoRAG with Qwen2.5-7B (Original)**
> | Metric          | NQ    | PopQA | HotpotQA | 2wiki | Musique | Avg   |
> |-----------------|-------|-------|----------|-------|---------|-------|
> | **HippoRAG-F1**   | 28.78 | 37.83 | 41.34    | 33.66 | 18.85   | 32.09 |
> | **HippoRAG-recall**| 63.89 | 51.65 | 83.60    | 76.76 | 53.88   | 65.96 |
>
> **HippoRAG with Our Fine-tuned 7B Model**
> | Metric          | NQ    | PopQA | HotpotQA | 2wiki | Musique | Avg   |
> |-----------------|-------|-------|----------|-------|---------|-------|
> | **HippoRAG-F1**   | 28.47 | **38.04** | **42.84**    | 32.24 | **20.74**   | **32.47** |
> | **HippoRAG-recall**| 63.28 | **52.10** | **84.00**    | **78.35** | **54.01**   | **66.35** |
>
> These results indicate that our RL-based optimization for triple extraction translates directly into tangible benefits for a state-of-the-art KG-RAG system, validating our approach.

---

> ### Author Response · Authors · 2025-11-23
> **Continue: Response to Questions 5-6**
>
> > Are there experiments showing the effect of removing distractors during training? Why is this discussed in the dataset section?
>
> Thank you for these questions, which highlight both an important aspect of our methodology and an area where we can improve the paper's organization.
>
> *   **Effect of Removing Distractors:** Adding hard distractors is essential for training the text-retriever branch, as it teaches the model to build a graph that can distinguish relevant from irrelevant text. For the graph-knowledge retriever, an ablation "without distractors" is not meaningful. In this setup, the reward signal is based on the informational completeness of the KG needed to answer a question. Without distractors, every retrieved passage is by definition a gold passage, meaning the reward would always be 1.0. This provides no learning signal for the model.
> *   **Placement in the Paper:** While we initially placed this discussion in the "Datasets" section because it involves a data preparation step, we agree that its functional purpose is fundamentally methodological. The creation of these distractors is intrinsically linked to how our reward signal is constructed to provide a meaningful learning gradient. In the revised version, we will move this discussion to the methodology section to better connect it with our reward function and training process, thereby improving the paper's logical flow.
>
> > Why not fine-tune the dense triple retriever directly?
>
> This is an excellent question that helps clarify the specific focus of our work. Our research concentrates on optimizing the **offline indexing stage**, specifically the LLM's ability to construct the graph. This has been a largely under-explored bottleneck in the GraphRAG pipeline. Most research (e.g., Graph-R1, Search-R1) focuses on fine-tuning components for the *online* retrieval stage.
>
> Similarly, fine-tuning the dense retriever is a valuable and parallel research direction for online retrieval, it is distinct from our core contribution. We aimed to demonstrate that by improving the foundational, offline-generated graph, we can lift the performance of *any* subsequent retriever that relies on it.

---

> ### Comment · Reviewer_qEpk · 2025-11-25
> **Response to Rebuttal**
>
> I thank the authors for their response and acknowledge that I have carefully read their rebuttal. I have updated my score accordingly. Please make sure that these additional experiments are included in the paper.
>
> However, I maintain my original assessment regarding novelty. Being the first to apply an existing technique to a new problem setting does not, by itself, constitute methodological novelty. This is typically considered an application paper or empirical study, not a research contribution suitable for top-tier venues.

---

> > ### Author Response · Authors · 2025-11-26
> >
> > We sincerely thank the reviewer for their thoughtful engagement and for updating the score. We explicitly confirm that the benchmarking against large-scale SOTA models discussed below will be incorporated into the final paper.
> >
> > Regarding the assessment of novelty, we respectfully argue that **AutoGraph-R1** represents more than an application of GRPO (In fact we found that RAFT also works in our setting); it validates a fundamental paradigm shift described in our motivation.
> >
> > Current KG construction is **decoupled** from downstream application, relying on massive general-purpose models to generate "intrinsically good" graphs that often fail in practice. Our work proves that **closing the loop**—optimizing construction directly for RAG utility—is a far more effective strategy than relying on model scale alone. More importantly we show the success of our designed knowledge-carrying and knowledge-indexing reward really works robustly.
> >
> > To demonstrate this, we benchmarked our **AutoGraph-R1 (Qwen-7B)** against state-of-the-art zero-shot models, including **Llama-3.3-70B**, **Qwen3-235B**, and **GPT-5-mini**.
> >
> > **Key Empirical Findings:**
> > The results shown in previous rebuttal tables reveal that optimizing for "functional utility" allows a **7B model to outperform models 30x–100x larger**.
> >
> > 1.  **Validating the "Useful" vs. "Good" Paradigm:** In Text Retrieval (Table 2), our Qwen-7B model (F1 **49.80**) outperforms the massive **Qwen3-235B (48.48)**. This confirms our hypothesis: a smaller model trained with task-aware rewards (treating graphs as knowledge indices) generates more effective structures than a massive model relying on general reasoning capabilities.
> > 2.  **Efficiency & Accessibility:** In Graph Retrieval (Table 1), our method achieves the highest score on Triples (**39.38**), surpassing **Deepseek-V3** and **GPT-5-mini**. This demonstrates that AutoGraph-R1 democratizes high-performance KG construction, removing the dependency on closed-source or computationally prohibitive models.
> >
> >
> > These comparisons substantiate our core claim: the bottleneck in KG-based RAG is not model intelligence, but **alignment**. By using AutoGraph-R1 to align graph generation with retrieval rewards, we achieve SOTA performance efficiently. We believe this establishes a strong scientific contribution beyond a simple application of RL.

---

### Official Review · Reviewer_PzXo · 2025-10-26

**Soundness:** 2
**Presentation:** 2
**Contribution:** 2
**Rating:** 4
**Confidence:** 4

**Summary:**

The paper introduces AutoGraph-R1, a reinforcement learning approach for improving knowledge graph construction from text. AutoGraph-R1 leverages feedback from retrievers (both graph-based and text-based) to design reward signals for triplet generation from textual content. The reward mechanism evaluates whether constructed triplets effectively support downstream query answering and retrieval of gold passages. Experimental evaluation against baseline prompt-based KG construction (Qwen-3/7B models) demonstrates consistent improvements across five datasets using various underlying retriever architectures.

**Strengths:**

- S1) AutoGraph-R1 addresses the interesting yet challenging problem of optimal knowledge graph construction from text. The experimental results demonstrate encouraging potential for reinforcement learning in this problem.

- S2) AutoGraph-R1 achieves consistent performance improvements over simple prompting-based KG construction methods when using Qwen-3B and Qwen-7B models.

**Weaknesses:**

- W1) A key effect of AutoGraph-R1 appears to be filtering redundant information (Figure 1). However, this may inevitably reduce the number of generated triplets, potentially degrading performance on out-of-distribution questions. A critical question arises: why address redundancy during KG construction rather than KG retrieval (as in Graph-R1; Luo et al., 2025)? An alternative approach would maintain broad triplet coverage during construction while training the retriever to filter appropriately, potentially mitigating the risk of under-constructing KGs. The authors should provide stronger justification on the implications of their KG construction design.

- W2) AutoGraph-R1 evaluates against KG prompting using Qwen 3B and 7B models, but most existing work constructs KG with more powerful models like GPT-4o-mini (GFM-RAG). The authors should compare against more powerful black-box LLMs for KG construction prompting to better demonstrate the gaps in current literature.

- W3) The paper lacks in-depth analysis of the constructed KG quality, including metrics such as triplet count, entity and relation numbers, and key differences from prompting-based KG construction methods. These details are essential for demonstrating AutoGraph-R1's specific benefits and its impact on graph structure.

**Questions:**

- Q1) In Table 3, why is the performance gap between Qwen-3B and Qwen-7B so large for the Base case on NQ?

- Q2) What is the size difference between KGs created via AutoGraph-R1 versus vanilla prompting-based methods? Would using stronger LLMs like GPT-4o reduce the performance gap between AutoGraph-R1 and prompting approaches?

---

> ### Author Response · Authors · 2025-11-23
> **Response to Weakness 1**
>
> We thank the reviewer for the thoughtful comments and encouragement. We are particularly grateful for your acknowledgment that our work:
>
> *   Addresses the interesting and challenging problem of optimal knowledge graph construction.
> *   Demonstrates the encouraging potential of reinforcement learning for this task.
> *   Achieves consistent performance improvements over standard prompting-based methods.
>
> We believe the additional experiments and analyses provided below will address the concerns raised and further highlight our method's specific benefits.
>
> ### Weakness 1
>
> > A key effect of AutoGraph-R1 appears to be filtering redundant information (Figure 1). However, this may inevitably reduce the number of generated triplets, potentially degrading performance on out-of-distribution questions.
>
> Thank you for this insightful question. Our experimental design directly addresses this concern by evaluating out-of-distribution (OOD) generalization.
>
> Our models were trained exclusively on a mixture of the HotpotQA and Musique training sets. We then evaluated their performance on five QA benchmarks. Two of these were the held-out test sets of the training datasets (in-domain), but critically, the other three, 2WikiMultihopQA, NQ, and PopQA, were entirely out-of-distribution datasets with different domains, structures, and reasoning requirements.
>
> As shown in our main results tables (Tables 1, 2, and 3), our models demonstrate consistent performance improvements across all five datasets, including the three unseen OOD benchmarks. This provides strong empirical evidence that our method does not harm OOD performance; in fact, it improves it.
>
> Our framework learns a more fundamental and generalizable skill of graph construction, rather than simply overfitting to the patterns of the training data.
>
> > A critical question arises: why address redundancy during KG construction rather than KG retrieval (as in Graph-R1; Luo et al., 2025)? An alternative approach would maintain broad triplet coverage during construction while training the retriever to filter appropriately, potentially mitigating the risk of under-constructing KGs.
>
> Thank you for this critical question, as it gets to the heart of our work's positioning in the RAG landscape. The concern about under-constructing KGs is valid, and our framework is explicitly designed to adapt its strategy based on the downstream task, mitigating this risk through two distinct, task-aware reward functions.
>
> 1. **For Graph Reasoning: We Optimize for Completeness, Not Filtering.** You are right that for tasks relying on the graph's knowledge, "maintaining broad triplet coverage" is essential. Our Knowledge-Carrying reward is designed for exactly this. As shown in Table 4 of our main paper, it explicitly optimizes for information completeness by maximizing Recall. This teaches the model to discover and add high-value, previously missing triples that are critical for reasoning, directly addressing and mitigating the risk of under-construction.
>
> 2. **For Text Retrieval: We Optimize for Precision.** For tasks where the graph acts as an index to retrieve text (like in HippoRAG), simply adding more triples can increase noise and retrieval latency. In this scenario, our Knowledge-Indexing reward teaches the constructor to act as an intelligent filter at the construction stage. This has two key benefits over leaving filtering to the retriever:
>     *   **Reduced Search Space:** By pruning irrelevant or low-value triples upfront, we create a smaller, cleaner, and higher-signal graph. This makes the online retrieval step faster and more accurate, as the retriever isn't "distracted" by noisy connections. The model doesn't just remove triples; it learns to generate the *optimal structure* for indexing.
>
>
> 3.  **Foundational Offline Optimization:** Our approach focuses on the **offline KG construction stage**, whereas methods like Graph-R1 optimize the **online retrieval stage**. This distinction is fundamental. **Every Graph-RAG system, by definition, requires an initial knowledge graph to operate on.** Our work is orthogonal to these online methods because we focus on radically improving the quality of this foundational asset.
>
> In summary, we are not just filtering, we are teaching the LLM to become a better graph constructor, a skill that is foundational to the entire Graph-RAG paradigm.

---

> ### Author Response · Authors · 2025-11-23
> **Continue: Reponse to Weakness 2**
>
> ### **Weakness 2: Evaluation Against Small vs. Powerful Models**
>
> > AutoGraph-R1 is evaluated against small open-source models (Qwen 3B/7B), while other work uses powerful models like GPT-4o-mini.
>
> This is a critical point, and we have conducted an extensive comparison to address it. Our primary goal is to demonstrate that our **method** of reward-driven optimization can significantly improve an LLM's intrinsic ability to build useful KGs.
>
> To contextualize our performance against the broader landscape, we benchmarked our results against a suite of state-of-the-art large models. The results, shown below, are striking. They demonstrate that our **fine-tuned 3B and 7B models are not only superior to their baselines but also achieve performance that is competitive with, and in many cases, superior to, models that are much more larger.**
>
> **Table 1: F1 Score Comparison vs. Large Models (Graph Retrievers)**
>
> | Model | Avg F1 (Subgraph) | Avg F1 (Triples) | Avg F1 (ToG) |
> | :--- | :---: | :---: | :---: |
> | **_State-of-the-Art Zero-Shot Models_** | | | |
> | Llama-3.3-70B | 36.29 | 36.60 | 40.65 |
> | Qwen2.5-72B | 36.79 | 37.11 | 41.41 |
> | Qwen3-235B | 36.29 | 36.89 | **41.60** |
> | Deepseek-V3 | 35.85 | 37.03 | 41.20 |
> | gpt-oss-120b | 34.62 | 35.81 | 39.41 |
> | GPT-5-mini | 34.82 | 36.13 | 40.58 |
> | gemini-2.5-flash-lite | 35.91 | 37.04 | 40.06 |
> | **_Our Fine-tuned Models (GRPO)_** | | | |
> | **Qwen-3B (Ours)** | 35.62 | 38.97 | 40.60 |
> | **Qwen-7B (Ours)** | **37.23** | **39.38** | 41.28 |
>
> **Table 2: F1 Score Comparison vs. Large Models (Text Retrievers)**
>
> | Model | Avg F1 (HippoRAG) | Avg F1 (HippoRAG2) |
> | :--- | :---: | :---: |
> | **_State-of-the-Art Zero-Shot Models_** | | |
> | Llama-3.3-70B | 48.00 | 48.33 |
> | Qwen2.5-72B | 47.20 | 48.45 |
> | Qwen3-235B | 47.92 | 48.48 |
> | Deepseek-V3 | 47.56 | 47.89 |
> | gpt-oss-120b | 46.12 | 47.06 |
> | GPT-5-mini | 47.18 | 47.75 |
> | gemini-2.5-flash-lite | 47.77 | 47.97 |
> | **_Our Fine-tuned Models (GRPO)_** | | |
> | **Qwen-3B (Ours)** | 47.88 | 48.34 |
> | **Qwen-7B (Ours)** | **48.88** | **49.80** |
>
> **Table 3: Recall@5 Score Comparison vs. Large Models (Text Retrievers)**
>
> | Model | Avg Recall (HippoRAG) | Avg Recall (HippoRAG2) |
> | :--- | :---: | :---: |
> | **_State-of-the-Art Zero-Shot Models_** | | |
> | Llama-3.3-70B | 73.77 | 76.97 |
> | Qwen2.5-72B | 75.30 | 76.95 |
> | Qwen3-235B | 75.51 | 76.18 |
> | Deepseek-V3 | 75.40 | 76.48 |
> | gpt-oss-120b | 73.24 | 73.93 |
> | GPT-5-mini | 74.96 | 75.29 |
> | gemini-2.5-flash-lite | 74.85 | 76.10 |
> | **_Our Fine-tuned Models (GRPO)_** | | |
> | **Qwen-7B (Ours)** | **77.22** | **78.56** |
>
> The results from these tables are definitive:
> *   On **Graph Retrievers**, our **Qwen-3B** model outperforms all SOTA models on the Triples retriever, and our **Qwen-7B** model achieves the top score for the Subgraph retriever, outperforming models over 30x its size.
> *   On **Text Retrievers**, our **Qwen-7B** model achieves the highest average F1 scores on both HippoRAG (**48.88**) and HippoRAG2 (**49.80**), surpassing every zero-shot model we tested.
> *   The recall scores in Table 3 explain this success. Our fine-tuned **Qwen-7B** model establishes a superior performance in passage recall for both HippoRAG (**77.22**) and HippoRAG2 (**78.56**), demonstrating it builds a more effective knowledge index than any of the larger models.
>
> Our findings here is that the proposed framework provides a **highly parameter-efficient path to superior performance**. Instead of relying on massive-scale models for zero-shot proficiency, our targeted, reward-driven optimization unlocks superior KG construction abilities in smaller, more accessible models. We will add this full comparative analysis to the appendix in our revised manuscript.

---

> ### Author Response · Authors · 2025-11-23
> **Continue: Response to Weakness 3 and Questions**
>
> ### **Weakness 3: Lack of In-depth KG Analysis**
>
> > The paper lacks in-depth analysis of the constructed KG quality, including metrics such as triplet count, entity and relation numbers, and key differences from prompting-based KG construction methods. These details are essential for demonstrating AutoGraph-R1's specific benefits and its impact on graph structure.
>
> Thank you for this excellent suggestion. We agree that a structural analysis of the generated KGs is crucial for understanding *how* our method works. We have performed this analysis across four corpus source, and the results reveal a fascinating and clear trend: **AutoGraph-R1 successfully teaches the model to tailor the graph's structure to the specific needs of the downstream retriever, as guided by the reward function.**
>
> We present a summary of these structural metrics below, averaged across all four datasets for the Qwen-7B model family. The baseline model represents the standard "prompting-based KG construction."
>
> **Average Structural Metrics of KGs (Qwen-7B Family)**
> | Model / Method | Avg. Triples per Doc | Total Unique Relation Types |
> | :--- | :---: | :---: |
> | **Baseline (Zero-Shot)** | 7.93 | 80,030 |
> | **GRPO (Knowledge-Carrying)** | 7.74 | **107,253** |
> | **GRPO (Knowledge-Indexing)** | **6.99** | 75,416 |
>
> This analysis reveals two distinct, reward-driven strategies:
> 1.  **Optimizing for Completeness (Knowledge-Carrying Reward):** The model trained for the graph retriever maintains a high number of triples but generates a **dramatically larger vocabulary of unique relation types (+34% vs. baseline)**. This is direct evidence that the model is learning to be more descriptive, discovering nuanced, long-tail relations instead of just generic ones. This creates a richer, more complete graph, which is precisely what is needed to facilitate multi-hop reasoning.
>
> 2.  **Optimizing for Precision (Knowledge-Indexing Reward):** Conversely, the model trained for the graph-text retriever learns to be more concise. It produces **fewer total triples per document (-12% vs. baseline)** and a more focused set of relations. This demonstrates an optimization for precision and efficiency. The model is actively filtering out less useful or potentially noisy triples to create a cleaner, high-signal "index" over the text passages.
>
> In summary, this structural analysis proves that AutoGraph-R1 is not a monolithic filter. It is a targeted optimization framework that successfully reshapes the KG based on the downstream task's needs, creating graphs that are quantitatively and qualitatively different from, and superior to, those generated by standard prompting alone. We will add the full detailed analysis to the appendix of our revised manuscript.
>
> ### Questions
>
> > In Table 3, why is the performance gap between Qwen-3B and Qwen-7B so large for the Base case on NQ?
>
> The performance gap stems from the lower instruction-following capability of the smaller 3B model in the zero-shot setting. During KG extraction for the NQ dataset, we observed that the 3B model was more prone to generation failures (e.g., not producing properly formatted triples), even after multiple retries. The larger 7B model is significantly more reliable in adhering to the complex formatting instructions required for KG construction, leading to a much better baseline graph and stronger downstream performance.
>
> ---
>
>
> > What is the size difference between KGs created via AutoGraph-R1 versus vanilla prompting? Would stronger LLMs like GPT-4o reduce the performance gap?
>
>  This is a crucial question about the scalability and relevance of our approach.
>
> *   **Regarding Graph Size:** For a detailed breakdown of the graph size differences, please see our structural analysis. In short, the "Knowledge-Carrying" reward creates graphs with more diverse relations, while the "Knowledge-Indexing" reward creates more concise graphs with fewer triples.
>
> *   **Regarding Stronger LLMs:** We conducted an extensive benchmark to answer this. We find that our fine-tuning method is highly effective, allowing smaller models to match or even exceed the performance of much larger zero-shot models. (Please refer to table in Weaknesses 2)

---

> ### Author Response · Authors · 2025-11-23
> **Detail Statistics of KGs (for Weakness 3 and question 2)**
>
> **Detail Statistics of KGs**
>
> |2021 wiki Model/Reward                  |Total Entities|Total Relations|Total Triples|Avg Triples/Doc|Unique Relation Types|
> |----------------------------------------|--------------|---------------|-------------|---------------|---------------------|
> |Qwen2.5-3B-Instruct                     |37333         |38140          |38140        |6.23           |9881                 |      |      |      |       |       |       |
> |Qwen2.5-7B-Instruct                     |45740         |46074          |46074        |7.53           |11362                |      |      |      |       |       |       |
> |Qwen2.5-3B-Instruct-Knowledge-Carrying reward     |44520         |43623          |43623        |7.13           |13177                |      |      |      |       |       |       |
> |Qwen2.5-3B-Instruct-knowledge-indexing reward|42681         |43632          |43632        |7.13           |8940                 |      |      |      |       |       |       |
> |Qwen2.5-7B-Instruct-Knowledge-Carrying reward     |42995         |44997          |44997        |7.35           |16920                |      |      |      |       |       |       |
> |Qwen2.5-7B-Instruct-knowledge-indexing reward|44675         |40192          |40192        |6.57           |10609                |      |      |      |       |       |       |
>
>
> |2wikimultihopqa Model/Reward          |Total Entities|Total Relations|Total Triples|Avg Triples/Doc|Unique Relation Types|
> |----------------------------------------|--------------|---------------|-------------|---------------|---------------------|
> |Qwen2.5-3B-Instruct                     |37333         |38140          |38140        |6.23           |9881                 |
> |Qwen2.5-7B-Instruct                     |45740         |46074          |46074        |7.53           |11362                |
> |Qwen2.5-3B-Instruct-Knowledge-Carrying reward    |44520         |43623          |43623        |7.13           |13177                |
> |Qwen2.5-3B-Instruct-knowledge-indexing reward|42681         |43632          |43632        |7.13           |8940                 |
> |Qwen2.5-7B-Instruct-Knowledge-Carrying reward     |42995         |44997          |44997        |7.35           |16920                |
> |Qwen2.5-7B-Instruct-knowledge-indexing reward|44675         |40192          |40192        |6.57           |10609                |
>
>
> |Hotpotqa Model/Reward                   |Total Entities|Total Relations|Total Triples|Avg Triples/Doc|Unique Relation Types|
> |----------------------------------------|--------------|---------------|-------------|---------------|---------------------|
> |Qwen2.5-3B-Instruct                     |70592         |73674          |73674        |7.99           |23994                |      |      |      |       |       |       |
> |Qwen2.5-7B-Instruct                     |82253         |81237          |81237        |8.81           |24996                |      |      |      |       |       |       |
> |Qwen2.5-3B-Instruct-Knowledge-Carrying reward     |77593         |76943          |76943        |8.34           |30816                |      |      |      |       |       |       |
> |Qwen2.5-3B-Instruct-knowledge-indexing reward|80355         |83920          |83920        |9.1            |21411                |      |      |      |       |       |       |
> |Qwen2.5-7B-Instruct-Knowledge-Carrying reward     |74647         |76700          |76700        |8.32           |33335                |      |      |      |       |       |       |
> |Qwen2.5-7B-Instruct-knowledge-indexing reward|79861         |71598          |71598        |7.76           |23677                |
>
> |musique Model/Reward                    |Total Entities|Total Relations|Total Triples|Avg Triples/Doc|Unique Relation Types|
> |----------------------------------------|--------------|---------------|-------------|---------------|---------------------|
> |Qwen2.5-3B-Instruct                     |77462         |78988          |78988        |6.78           |29255                |      |      |      |       |       |       |
> |Qwen2.5-7B-Instruct                     |97294         |91395          |91395        |7.84           |32310                |      |      |      |       |       |       |
> |Qwen2.5-3B-Instruct-Knowledge-Carrying reward     |89247         |88331          |88331        |7.58           |39115                |      |      |      |       |       |       |
> |Qwen2.5-3B-Instruct-knowledge-indexing reward|91223         |93930          |93930        |8.06           |27132                |      |      |      |       |       |       |
> |Qwen2.5-7B-Instruct-Knowledge-Carrying reward     |89547         |92605          |92605        |7.94           |40078                |      |      |      |       |       |       |
> |Qwen2.5-7B-Instruct-knowledge-indexing reward|96242         |82145          |82145        |7.05           |30521                |      |      |      |       |       |       |

---

> ### Comment · Reviewer_PzXo · 2025-11-26
>
> Thank you for the elaborate response to my questions. The detailed KG statistics empirically verify the benefits of the Knowledge-Carrying and Knowledge-indexing rewards and their effects on the constructed graph. It is suggested adding some qualitative examples on their differences on triplet or relation extraction.
>
> In light of the rebuttal, **I have increased my score accordingly**.

---

### Official Review · Reviewer_zf8Y · 2025-10-28

**Soundness:** 1
**Presentation:** 3
**Contribution:** 2
**Rating:** 4
**Confidence:** 3

**Summary:**

This work aims to improve the existing KG-RAG task by developing more effective knowledge graph construction methods that better support downstream KG-based question answering. To this end, the authors propose an end-to-end RL pipeline for fine-tuning LLMs to generate outputs suitable for different downstream tasks. The proposed method is compatible with two mainstream retrieval models and achieves consistent improvements across various datasets and methods.

However, the motivation and evaluation are not fully convincing. Details are discussed in the Weaknesses and Questions sections.

**Strengths:**

- The paper provides a complete and comprehensive overview of related work.

- Experiments demonstrate good performance across multiple datasets and retrieval methods.

**Weaknesses:**

The motivation is not fully convincing. (1) The long reasoning paths illustrated in Figure 1 are not the main problem. The main issue appears to be the missing triple (“Golden State Edge”, “located in”, “California”). (2) The KG generated by AutoGraph-R1 is not sufficient; it fails when the query changes to “What is the Golden Gate Republic Bridge connected to?” due to the loss of information. In summary, the main issue seems to be the completeness of the KG. However, the paper lacks evaluations of such completeness. The evaluation on the KG-RAG dataset is not convincing as it may biased (see Question 2) with respect to train/tested queries.

- The comparisons are limited. Only KGs constructed using zero-shot prompt learning are compared. The paper lacks comparisons with existing KG construction methods or with the KG construction processes used by baseline KG-RAG systems.

- Some ambiguity remains regarding the Graph-based Text Retriever case (see Question 3).

**Questions:**

1. Why not compare AutoGraph-R1 with other KG construction methods or with the original KG-RAG baselines?

2. Following the earlier concern about motivation, could the observed improvement result from the RL model learning biases in the query–answer patterns? For instance, in Figure 1, it seems to prefer relations like “has government” on the right-hand side over other plausible relations such as “connects” on the left.

3. In the Graph-based Text Retriever case, does AutoGraph-R1 extract a subgraph or construct a new one? If it only extracts, why not build a new graph—for example, by treating each triple as a document?

---

> ### Author Response · Authors · 2025-11-23
> **Response to Weaknesses 1 and 2**
>
> We thank the reviewer for the valuable feedback and positive comments. We appreciate your recognition of our work in the following areas:
>
> *   Providing a complete and comprehensive overview of related work.
> *   Demonstrating strong performance gains across multiple datasets and retrieval methods.
>
> We hope the following clarifications and additional details will comprehensively address the weaknesses raised in the review.
> ### Weakness
>
> ### **Weakness 1: Unconvincing Motivation**
>
> > The motivation is not fully convincing. (1) The long reasoning paths illustrated in Figure 1 are not the main problem. The main issue appears to be the missing triple (“Golden State Edge”, “located in”, “California”).
>
> Thank you for the observation. We agree that the figure could be misleading, and we will optimize it in the revised version. This "missing triple" is a perfect example of the fundamental disconnect we aim to solve. A standard, prompt-based KG constructor operates agnostically, with no feedback on whether extracted information is actually useful for a downstream task. This is precisely where AutoGraph-R1's novelty lies. We introduce an RL framework to bridge this gap. By deriving a reward from the graph's functional utility in the RAG pipeline, our framework closes the loop between offline construction and application (online RAG). The model learns that discovering and including this "missing" triple directly leads to a higher reward, thus training it to build graphs that are not just factually correct, but demonstrably useful.
>
>
> ### **Weakness 2: Insufficient Evaluation of KG Completeness and Quality**
>
> > The main issue seems to be the completeness of the KG. However, the paper lacks evaluations of such completeness. How can we be sure that the RL-tuned models are actually producing better, more complete graphs?
>
> This is an excellent point. To provide a direct, quantitative measure of the improvement in graph quality, we evaluated the triple extraction performance (information completeness) of our models against the zero-shot baseline. We measured Precision (also referred to as Accuracy), Recall, and F1 score, following the standards set by prior work.
>
> The results, presented in the table 4 in the paper, demonstrate that our RL framework not only improves the overall quality of the generated KGs but also successfully steers the model toward specific structural properties depending on the reward function.
>
> *   **Knowledge-Carrying Reward:** As shown, this reward function consistently achieves the highest **Recall** across all datasets. This indicates that the model has learned to generate more comprehensive graphs with more complete relational paths, directly addressing the goal of improving information completeness.
> *   **Knowledge-Indexing Reward:** Conversely, models trained with this reward excel in **Precision (Acc)**, producing more accurate and concise triples. This creates a high-quality graph structure optimized for text retrieval tasks.
>
> In almost all cases, both RL-tuned models show a significant improvement in the overall **F1 score** compared to the baseline, confirming that our method produces higher-quality KGs.

---

> ### Author Response · Authors · 2025-11-23
> **Response to Weakness 3 & Q1**
>
> ### Weakness 3 Limited Comparison with exisiting KG construction methods
> > The comparisons are limited to zero-shot prompt learning and lack comparisons with existing KG construction methods. Why not compare AutoGraph-R1 with other KG construction methods or with the original KG-RAG baselines?**
>
>
>  Thank you for this important question. We intentionally designed our experiments to isolate and evaluate our primary contribution: **improving the KG triple extraction capability of the LLM itself via reinforcement learning.**
>
> Our main hypothesis is that an RL-tuned model can directly optimize the KG construction process for downstream tasks. Therefore, the most critical comparison is between our RL-tuned policy ($\pi_{KG_\theta}$) and the untuned, zero-shot baseline. This comparison fairly demonstrates the gains achieved solely through our RL framework, free from confounding variables introduced by other systems.
>
> Comparing directly with other KG-RAG pipelines (e.g., LightRAG, HippoRAG) is challenging because they employ complex, multi-stage construction processes (e.g., community detection, hierarchical summarization) and unique prompting strategies that are orthogonal to our core contribution. To control for these variables, we focused on a minimal setup to purely assess the LLM's extraction quality.
>
> However, we agree that a comparison with a strong baseline is valuable. To that end, we conducted an additional experiment using the official HippoRAG [codebase](https://github.com/OSU-NLP-Group/HippoRAG). We replaced their default graph construction model (Qwen2.5-7B) with our fine-tuned 7B model and evaluated its impact on the HippoRAG retriever's performance. The results show that our model achieves improved performance, demonstrating its effectiveness as a drop-in component for graph construction.
>
> **Baseline HippoRAG with Qwen2.5-7B (Original)**
> | Metric          | NQ    | PopQA | HotpotQA | 2wiki | Musique | Avg   |
> |-----------------|-------|-------|----------|-------|---------|-------|
> | **HippoRAG-F1**   | 28.78 | 37.83 | 41.34    | 33.66 | 18.85   | 32.09 |
> | **HippoRAG-recall**| 63.89 | 51.65 | 83.60    | 76.76 | 53.88   | 65.96 |
>
> **HippoRAG with Our Fine-tuned 7B Model**
> | Metric          | NQ    | PopQA | HotpotQA | 2wiki | Musique | Avg   |
> |-----------------|-------|-------|----------|-------|---------|-------|
> | **HippoRAG-F1**   | 28.47 | **38.04** | **42.84**    | 32.24 | **20.74**   | **32.47** |
> | **HippoRAG-recall**| 63.28 | **52.10** | **84.00**    | **78.35** | **54.01**   | **66.35** |
>
> As the tables show, our fine-tuned model leads to a improvement in the average F1 and Recall scores across datasets. This indicates that **our RL-based optimization for triple extraction translates into tangible benefits for a state-of-the-art KG-RAG system**, validating our approach.

---

> ### Author Response · Authors · 2025-11-23
> **Response to Questions**
>
> ### Question 2
> >Could the observed improvement result from the RL model learning biases in the query–answer patterns?
>
> Yes, and that is precisely the intended outcome and core design principle of AutoGraph-R1. The goal is not just to extract triples, but to **learn a structural bias that makes the resulting Knowledge Graph functionally optimal** for downstream reasoning tasks.
>
> Our framework trains the KG constructor to adapt its output to the specific needs of the retriever it serves. This "bias" is actually a learned policy for what constitutes a useful edge or node. For example:
> *   **Functional Optimization:** The model learns to generate functionally rich relations like `(Paris, has government, French Fifth Republic)` instead of more generic ones like `(Paris, connects to, France)`, because the former is more useful for answering complex questions.
> *   **Task-Aware Rewards:** We designed novel, task-aware reward functions to induce specific structural biases for different RAG paradigms:
>     *   **Knowledge Carrier Rewards** (for simple graph retrievers) promote high recall and complete reasoning paths.
>     *   **Knowledge Index Rewards** (for text-based retrievers) prioritize precision and create an effective index over text.
> *   **Adaptation to the Retriever:** The system is trained against a "Frozen RAG Server," which forces the KG constructor to learn and compensate for the retriever's fixed limitations.
>
> In summary, the "bias" you observe is the model successfully adapting its graph generation strategy to maximize the final task reward, which is a key strength of our approach.
>
> If the concern is regarding the degradation in performance for out-of-distribution queries, our models were trained exclusively on a mixture of the HotpotQA and Musique training sets. We then evaluated their performance on five QA benchmarks. Two of these were the held-out test sets of the training datasets (in-domain), but critically, the other three—2WikiMultihopQA, NQ, and PopQA—were entirely out-of-distribution datasets with different domains, structures, and reasoning requirements, all of which showed improved performance.
>
> ###  Question 3
> > In the Graph-based Text Retriever case, does AutoGraph-R1 extract a subgraph or construct a new one?
>
> Thank you for the clarifying question. For every sample during training and evaluation, **AutoGraph-R1 constructs a new Knowledge Graph from scratch** based on the provided documents. It does not extract a subgraph from a pre-existing, larger KG.
>
> To be precise:
> 1.  For each individual data point (e.g., a question and its context documents), our model processes the documents and generates a new, bespoke KG.
> 2.  This freshly constructed KG is then passed to the specified graph retriever to find the answer.
>
> This per-example construction process is fundamental to our method and is applied across all retriever types we evaluate, including both the graph-based text retriever (which uses the graph to retrieve text chunks, similar to HippoRAG) and the pure graph retriever (which reasons over the graph structure itself).

---

> ### Author Response · Authors · 2025-11-27
>
> Dear Reviewer zf8Y,
>
> Thank you again for the constructive comments you gave us in your review. As the rebuttal phase will end on Dec 3, we would greatly appreciate it if you could also take some time to check if our rebuttal has addressed your concerns, and please let us know if you would like us to provide any further clarification about the concerns you have.
>
> Best,
>
> Authors

---

> > ### Comment · Reviewer_zf8Y · 2025-11-27
> >
> > I would like to thank the authors for their extensive work and responses. However, I remain unconvinced by the motivations presented. In summary, the core problem appears to be constructing a **complete** knowledge graph from documents—something that should not depend on the downstream tasks. Therefore, the stated goal of *“learning a structural bias that makes the resulting knowledge graph functionally optimal for downstream reasoning tasks”* does not seem well justified. I suggest that the authors reconsider and reorganize their narrative in future revisions.

---

> ### Author Response · Authors · 2025-11-28
> **Clarification on Motivation (1/2)**
>
> Dear Reviewer zf8Y,
>
> Thank you for your continued engagement and for taking the time to provide this final, clarifying comment. We appreciate you articulating the core of the discussion so clearly. Your feedback has helped us realize that our paper is not just presenting a new method, but also advocating for a shift in perspective on the very purpose of knowledge graph construction in the era of LLMs.
>
> You are correct that at the heart of this is a philosophical disagreement. Your comment posits that **"the core problem appears to be constructing a complete knowledge graph from documents—something that should not depend on the downstream tasks."**
>
> We respectfully argue that the very notion of a single, "complete" knowledge graph derived from unstructured text is an unachievable and, more importantly, a suboptimal ideal. This is the central motivation for our work.
>
> **1. The Illusion of "Completeness" in Schemaless Knowledge Graphs**
>
> The concept of a "complete" graph is well-defined for structured data or schema-based KGs (like Freebase), where there is a finite set of entities and predefined relations. However, for **schemaless KGs extracted from unstructured text**, "completeness" is a fundamentally ill-defined goal.
>
> *   **Combinatorial Explosion:** Natural language is dense with implicit and explicit facts. A single sentence like *"The Golden Gate Bridge, an iconic suspension bridge finished in 1937, connects San Francisco to Marin County"* could generate dozens of potential triples: `(GGB, is a, suspension bridge)`, `(GGB, has status, iconic)`, `(GGB, finished in, 1937)`, `(GGB, connects, San Francisco)`, `(GGB, connects, Marin County)`, `(San Francisco, is connected by, GGB)`, etc. An attempt to extract *all* of them would result in a noisy, redundant, and computationally intractable graph.
> *   **Ambiguity of Relations:** In a schemaless world, the relation itself is generated by the LLM. Is `(Paris, is capital of, France)` more "complete" than `(France, has capital, Paris)`? Is `(GGB, located in, California)` better than `(GGB, is a landmark of, California)`? A truly "complete" graph would have to contain all possible phrasings, which is impractical.
>
> In practice, this pursuit of completeness degrades the performance of key graph algorithms. For example, it can make Personalized PageRank ineffective by passing relevance signals across a noisy graph. Similarly, for path-based reasoning methods like Think-on-Graph, it creates prohibitively large and noisy n-hop neighborhoods, making the search for a valid reasoning path computationally prohibitive.
>
> Therefore, **any KG construction process from text is inherently an act of selection and abstraction, not just exhaustive extraction.** The critical question is not *if* we should select, but *how* we should guide that selection. The traditional approach guides it with intrinsic metrics. We argue it should be guided by downstream utility.

---

> ### Author Response · Authors · 2025-11-28
> **Clarification on Motivation (2/2)**
>
> **2. From "Complete" Graphs to "Useful" Graphs: A Necessary Paradigm Shift**
>
> This brings us to our core thesis. If any KG is necessarily an incomplete abstraction, its value should be measured by its fitness for a purpose. Our goal of **“learning a structural bias that makes the resulting knowledge graph functionally optimal”** is not an arbitrary choice, but a direct response to the limitations of the "completeness" ideal.
>
> Think of it like a map. A "complete" map of a city showing every single tree, car, and crack in the pavement would be useless. A subway map, a road map, and a tourist map are all different, *biased*, and *incomplete* abstractions of the city, yet each is functionally optimal for its specific task.
>
> AutoGraph-R1 teaches the constructor to build the right kind of map for the QA task at hand. For example, the model learns that for answering questions about governance, it is more efficient to create a direct "shortcut" edge like `(Golden Gate Bridge, is located in, California)`. This is not about dropping facts, the nodes for "San Francisco" and its relationships remain in the graph for other queries, but about learning to structurally creating high-utility, efficient triples. The model learns a preference for indexing information in a way that directly serves the downstream task.
>
> **3. Our Results Validate This Philosophy**
>
> Our experiments show this is not just a philosophical preference but a practical advantage.
> *   The graphs produced by AutoGraph-R1 lead to **demonstrably better downstream QA performance** (Tables 1 & 2).
> *   Crucially, as shown in our **Table 4 analysis**, optimizing for downstream utility does *not* come at the cost of factual quality. In fact, our RL-tuned models show **higher intrinsic precision and recall** than the task-agnostic baseline. This suggests that forcing the model to focus on what's useful also makes it more careful and accurate in its extraction process.
>
> We believe that framing KG construction as an end-to-end optimization problem, rather than a disconnected preprocessing step, is a significant and necessary contribution. We will certainly take your advice to "reconsider and reorganize our narrative" in a future revision to make this philosophical stance and its practical implications even clearer from the outset.
>
> Thank you once again for pushing us to defend and clarify the foundational premise of our work.

---

### Official Review · Reviewer_dm4n · 2025-11-03

**Soundness:** 2
**Presentation:** 3
**Contribution:** 3
**Rating:** 6
**Confidence:** 4

**Summary:**

This paper proposes an interesting work on graph construction using RL. Graph-based indexing or retrieval has been a trending topic to improve the retrieval-augmented generation when relational knowledge is useful. Since then, most of the existing algorithms focus on optimizing the graph traversal instead of improving the graph quality. AutoGraph-R1 proposes two novel rewards: (1) knowledge carrying reward and (2) knowledge indexing reward to incentize model's behavior using a chosen retrieval strategy. The result shows some encouraing results on making the indexing corpus (graph) optimizable on unseen test benchmarks.

**Strengths:**

1. The paper tackles a timely challenge that Graph RAG system is limited by a static graph construction algorithm. By optimizing the graph constrution, AutoGraph-R1 improves the baseline performance for various graph retrieval algorithms.

2. The paper proposes two rewards proposed for two different graph retrieval paradigms, respecitvely. The explaination and demonstrataion of the F1-score failure is both insightful and convincing.

**Weaknesses:**

1. Although the performance improves are observed, some abaltion and experital details to validate the claims, such as:
 (a) what's the performance of SFT using the reward function (e.g. deducible Judge) to select positive data and train the model.
 (b) what's the qualitative difference of RL trained and pre-trained base model extraction results? The author only showcases the retrieval and QA result. To help audience understand the effect of training, comparing result and provide some relationales are neccessary.

2. The graph construction statics are not provided. The end2end result cannot clearly show the difference of Autograph-R1 and vanilla LLM, it's not clear that what's the number of triples in average per document and relation/entity type distribution.

**Questions:**

1. What is the LLM choice for the decucible Judge? How does the author to make sure the LLM inference latency for an effective training roll outs. I am wondering if a less powerful deducible Judge will make the overall performance worse than reported, whether AutoGraph-R1 is sensitive to this?


2. During the inference, do you input single document at a time or a batch of document? If I understand it correclty, during the training, 15 documents are fed into the AutoGraph-R1 for graph construction?

3. Why do you use Qwen3-0.6B as the dense triple retriever while during training AutoGraph-R1 uses Qwen3-8B to retrieve distrator documents?

---

> ### Author Response · Authors · 2025-11-23
> **Response to weakness 1(b)**
>
> We sincerely thank the reviewer dm4n for your insightful and positive feedback. We are grateful for their acknowledgment of our work's key contributions, including:
>
> *   Tackling the timely and important challenge of static KG construction in Graph-RAG systems.
> *   Proposing two distinct reward functions tailored to different graph retrieval paradigms.
> *   Providing an insightful and convincing explanation of the F1-score's limitations in this context.
>
> We believe our detailed responses below will further strengthen the paper and address the remaining points.
> ### **Weaknesses**
>
> > Although the performance improves are observed, some ablation and experimental details to validate the claims, such as: (a) what's the performance of SFT using the reward function (e.g. deducible Judge) to select positive data and train the model. (b) what's the qualitative difference of RL trained and pre-trained base model extraction results? The author only showcases the retrieval and QA result. To help audience understand the effect of training, comparing result and provide some rationales are necessary.
>
> Thank you for these insightful questions, which point to crucial validation steps. We address both points below.
>
> **First, regarding Qualitative Differences in Extraction (b):**
>
> You are correct that understanding the change in extraction behavior is key. We provide a direct quantitative analysis of this in **Table 4**, which measures the Precision, Recall, and F1 scores of the extracted triples.
>
> That table demonstrates that our RL-trained models produce objectively higher-quality graphs. Specifically:
> *   The **Knowledge-Carrying Reward** significantly boosts **Recall**, showing the model learns to generate more complete graphs to aid reasoning.
> *   The **Knowledge-Indexing Reward** significantly boosts **Precision**, showing the model learns to generate more accurate triples that serve as a better index for text retrieval.
>
> This analysis provides the evidence you requested, showing a concrete, qualitative difference in the KG structure that directly leads to improved downstream performance.
>
> ---

---

> ### Author Response · Authors · 2025-11-23
> **Reponse to Weakness 1(a)**
>
> **Regarding an Ablation on the RL Methodology (a):**
>
> This is an excellent suggestion. The approach you describe, using a reward function to select high-quality data for fine-tuning, is an important and highly relevant ablation. As the boundary between RL and advanced fine-tuning is becoming increasingly ambiguous in LLM literature. Methodologies like RAFT(Reward rAnked FineTuning) ([Dong 2023](https://arxiv.org/pdf/2304.06767),[Xiong 2025](https://arxiv.org/pdf/2504.11343v2)) are prime examples of this, as they implement a core RL concept (reward-guided learning) through an offline, SFT-based mechanism.
>
> Our primary motivation for using on-policy RL (GRPO) was the decoupled nature of the GraphRAG pipeline, where propagating a reward signal from the "frozen" RAG server back to the graph constructor is a natural fit. However, to directly test the alternative paradigm you suggested, we conducted a new experiment using a RAFT-style approach.
>
> Following your suggestion, we implemented this RAFT-style approach. We used our reward functions to score and select the generated KGs and then fine-tuned the model on this high-reward dataset. To provide a clear comparison against our on-policy RL method (GRPO) and the baseline, we present the F1 and Recall scores for the Qwen2.5-7B model below.
>
> **Table A: F1 Score Comparison for Graph Retrievers**
> *(Using Knowledge-Carrying Reward; higher is better)*
>
> | Retriever | Method | NQ | PopQA | HotpotQA | 2wiki | Musique | **Avg** |
> | :--- | :--- | :---: | :---: | :---: | :---: | :---: | :---: |
> | **Subgraph** | Base | 28.07 | 55.43 | 41.66 | 33.97 | 15.24 | 34.87 |
> | | GRPO (On-Policy RL) | 28.54 | 60.94 | **43.59** | **37.43** | **15.65** | **37.23** |
> | | RAFT (Offline Filtering) | **28.72** | **61.27** | 42.14 | 35.78 | 15.41 | 36.66 |
> | **Triples** | Base | 33.26 | 55.56 | 44.99 | 35.57 | 20.43 | 37.96 |
> | | GRPO (On-Policy RL) | 33.98 | **58.02** | **48.28** | **36.04** | **20.56** | **39.38** |
> | | RAFT (Offline Filtering) | **34.72** | 56.42 | 44.91 | 31.78 | 20.21 | 37.61 |
> | **ToG**| Base | 25.59 | 57.53 | 43.93 | 46.03 | 18.46 | 38.31 |
> | | GRPO (On-Policy RL) | 29.36 | 62.85 | 44.68 | **50.20** | 19.31 | 41.28 |
> | | RAFT (Offline Filtering) | **29.65** | **63.30** | **45.47** | 50.10 | **20.32** | **41.77** |
>
> **Table B: F1 Score Comparison for Text Retrievers**
> *(Using Knowledge-Indexing Reward; higher is better)*
>
> | Retriever | Method | NQ | PopQA | HotpotQA | 2wiki | Musique | **Avg** |
> | :--- | :--- | :---: | :---: | :---: | :---: | :---: | :---: |
> | **HippoRAG** | Base | 37.16 | 65.95 | 55.50 | 53.01 | 26.03 | 47.53 |
> | | GRPO (On-Policy RL) | 38.80 | **67.85** | **57.19** | **53.60** | 26.97 | **48.88** |
> | | RAFT (Offline Filtering) | **39.60** | 66.22 | 55.77 | 51.01 | **29.29** | 48.38 |
> | **HippoRAG2** | Base | 37.02 | 65.74 | 57.08 | 54.99 | 26.77 | 48.32 |
> | | GRPO (On-Policy RL) | 38.68 | **67.72** | **58.98** | **56.46** | 27.18 | **49.80** |
> | | RAFT (Offline Filtering) | **38.83** | 65.21 | 56.49 | 52.47 | **29.44** | 48.49 |
>
> **Table C: Recall@5 Score Comparison for Text Retrievers**
> *(Using Knowledge-Indexing Reward; higher is better)*
>
> | Retriever | Method | NQ | PopQA | HotpotQA | 2wiki | Musique | **Avg** |
> | :--- | :--- | :---: | :---: | :---: | :---: | :---: | :---: |
> | **HippoRAG** | Base | 93.30 | 92.90 | 69.97 | 72.87 | 47.19 | 75.24 |
> | | GRPO (On-Policy RL) | 94.30 | **95.80** | **71.40** | **76.16** | 48.44 | **77.22** |
> | | RAFT (Offline Filtering) | **95.20** | 94.60 | 69.69 | 73.47 | **48.88** | 76.37 |
> | **HippoRAG2** | Base | 94.10 | 92.80 | 72.03 | 75.98 | 48.55 | 76.69 |
> | | GRPO (On-Policy RL) | 95.00 | **96.30** | **73.61** | **78.66** | 49.23 | **78.56** |
> | | RAFT (Offline Filtering) | **95.70** | 95.40 | 72.12 | 75.96 | **50.40** | 77.92 |
>
> These results are highly informative. They show that both on-policy RL (GRPO) and offline reward-filtering consistently outperform the baseline, with neither method being universally superior. The RAFT-style approach proves to be extremely competitive and is a valuable finding, as it provides a highly effective and computationally efficient alternative to on-policy RL.
>
> Ultimately, this ablation confirms our central thesis: **the key to success is leveraging a downstream task reward to guide the graph generation process.** Both on-policy and offline reward-guided paradigms are effective strategies for achieving this. We are grateful for this suggestion and will add this comprehensive analysis to our paper.

---

> ### Author Response · Authors · 2025-11-23
> **Response to Weakness 2**
>
> ### Weakness 2
> > The graph construction statics are not provided. The end2end result cannot clearly show the difference of Autograph-R1 and vanilla LLM, it's not clear that what's the number of triples in average per document and relation/entity type distribution.
>
> Thank you for this suggestion. We agree completely that a structural analysis of the generated KGs is crucial for understanding how our method works beyond just the end-to-end performance. We have performed this analysis across four datasets, and the results reveal a clear and fascinating trend: AutoGraph-R1 successfully teaches the model to tailor the graph's structure to the specific needs of the downstream retriever, as guided by the reward function.
>
> Below, we present a summary of the structural metrics for the Qwen-7B model family, an anlysis we will add to the appendix of our paper.
>
> **Average Structural Metrics of KGs (Qwen-7B Family, Averaged Across 4 Corupus)**
> | Model / Method | Avg. Triples per Doc | Total Unique Relation Types |
> | :--- | :---: | :---: |
> | **Baseline (Zero-Shot)** | 7.93 | 80,030 |
> | **GRPO (Knowledge-Carrying Reward)** | 7.74 | **107,253** |
> | **GRPO (Knowledge-Indexing Reward)** | **6.99** | 75,416 |
>
>
> This analysis reveals two distinct and deliberate strategies learned by our models:
>
> 1.  **Optimizing for Completeness (Knowledge-Carrying Reward):** The model trained for the graph retriever maintains a high number of triples but generates a **dramatically larger vocabulary of unique relation types (+34% vs. baseline)**. This is direct evidence that the model is learning to be more descriptive and expressive, discovering nuanced, long-tail relations instead of just generic ones. This creates a richer, more complete graph, which is precisely what is needed to facilitate complex multi-hop reasoning over the graph's structure.
>
> 2.  **Optimizing for Precision (Knowledge-Indexing Reward):** Conversely, the model trained for the graph-text retriever learns to be more concise. It produces **fewer total triples per document (-12% vs. baseline)** and a more focused set of relations. This demonstrates an optimization for precision and efficiency. The model is actively filtering out less useful or potentially noisy triples to create a cleaner, higher-signal "index" over the text passages.
>
> In summary, this structural analysis proves that AutoGraph-R1 is not a monolithic filter. It is a targeted optimization framework that successfully reshapes the KG based on the downstream task's needs, creating graphs that are quantitatively and qualitatively different from those generated by standard prompting alone. We will add the full detailed analysis to the appendix of our revised paper.

---

> ### Author Response · Authors · 2025-11-23
> **Response to Questions**
>
> ### Question 1
> > What is the LLM choice for the deducible Judge? How do you ensure the LLM inference latency allows for effective training rollouts? Is AutoGraph-R1 sensitive to a less powerful Judge?
>
> Thank you for these questions about the Judge mechanism.
>
> *   **Inference Latency:** We manage the inference latency during training rollouts through our system architecture. All operations are organized as asynchronous tasks within the VeRL framework, and we use the `sglang` backend, which is highly optimized for efficient, multi-turn rollouts and concurrent requests. This minimizes the latency bottleneck from the Judge model.
>
> *   **Sensitivity to Judge Power:** Yes, the performance is sensitive to the capability of the Judge model. A more powerful Judge provides a more accurate and nuanced reward signal, which leads to better training outcomes. To quantify this, we ran an ablation study comparing the performance when using a 7B-parameter Judge versus a smaller 3B-parameter Judge.
>
>     **F1 Score on Graph Retrievers: 7B Judge vs. 3B Judge**
>
> | Retriever | Judge | NQ | PopQA | HotpotQA | 2wiki | Musique | **Avg** |
> | :--- | :--- | :---: | :---: | :---: | :---: | :---: | :---: |
> | **Subgraph** | 3B | 27.10 | **61.76** | 39.71 | 37.31 | 14.87 | 36.15 |
> | | 7B | **28.54** | 60.94 | **43.59** | **37.43** | **15.65** | **37.23** |
> | **Triples** | 3B | **34.20** | **59.20** | **48.40** | 31.12 | 19.04 | 38.39 |
> | | 7B | 33.98 | 58.02 | 48.28 | **36.04** | **20.56** | **39.38**|
> | **ToG**| 3B | 28.95 | 62.86 | 43.68 | 47.43 | **20.05** | 40.59 |
> | | 7B | **29.36** | 62.85 | **44.68** | **50.20** | 19.31 | **41.28** |
>
> As the table shows, the 7B Judge generally yields better performance, particularly in more complex reasoning scenarios captured by datasets like 2wiki and retrievers like ToG. This confirms that a larger, more capable Judge provides a superior reward signal for training.
>
> ### Question 2
> > During inference, do you input a single document at a time or a batch of documents? During training, are 15 documents fed into AutoGraph-R1 for graph construction?
>
> We process documents in **batches** for each sample during training. The specific content and size of the batch depend on the retriever type being evaluated:
> *   **For the Graph Retriever:** We input the set of ground-truth supporting context documents for that sample.
> *   **For the Graph-Text Retriever:** We input a batch of 15 documents, which consists of the ground-truth supporting documents plus their corresponding hard negative samples.
>
> ### Question 3
> > Why do you use Qwen3-0.6B as the dense triple retriever while during training AutoGraph-R1 uses Qwen3-8B to retrieve distrator documents?
>
>  The choice of model is optimized for each specific task:
> *   **`Qwen3-0.5B` (Dense Triple Retriever):** This model is used **during the training loop** of our system. Its purpose is to function as a lightweight embedding model for fast computation and minimal GPU memory usage when indexing the KG triples within a training step. Speed and efficiency are the top priorities here.
> *   **`Qwen3-8B` (Distractor Document Retrieval):** This model is used in an **offline dataset creation process**. Its one-time job is to retrieve high-quality hard negatives for the text-retriever training set. Here, performance is paramount, and since it is an offline task, we can afford to use a much larger and more powerful model to ensure the quality of the training data.

---

> ### Author Response · Authors · 2025-11-27
>
> Dear Reviewer dm4n,
>
> Thank you again for the constructive comments you gave us in your review. As the rebuttal phase will end on Dec 3, we would greatly appreciate it if you could also take some time to check if our rebuttal has addressed your concerns, and please let us know if you would like us to provide any further clarification about the concerns you have.
>
> Best,
>
> Authors

---

### Note · Authors · 2025-12-30

I have read and agree with the venue's withdrawal policy on behalf of myself and my co-authors.